# The Clock and the Pizza: Two Stories in Mechanistic Explanation of Neural Networks

**Ziqian Zhong\*, Ziming Liu\*, Max Tegmark, Jacob Andreas**
Massachusetts Institute of Technology
{ziqianz, zmliu, tegmark, jda}@mit.edu

## Abstract

Do neural networks, trained on well-understood algorithmic tasks, reliably re-discover known algorithms for solving those tasks? Several recent studies, on tasks ranging from group arithmetic to in-context linear regression, have suggested that the answer is yes. Using modular addition as a prototypical problem, we show that algorithm discovery in neural networks is sometimes more complex. Small changes to model hyperparameters and initializations can induce discovery of qualitatively different algorithms from a fixed training set, and even parallel implementations of multiple such algorithms. Some networks trained to perform modular addition implement a familiar *Clock* algorithm (previously described by Nanda et al. [1]); others implement a previously undescribed, less intuitive, but comprehensible procedure we term the *Pizza* algorithm, or a variety of even more complex procedures. Our results show that even simple learning problems can admit a surprising diversity of solutions, motivating the development of new tools for characterizing the behavior of neural networks across their algorithmic phase space. [1]

## 1 Introduction

Mechanistically understanding deep network models—reverse-engineering their learned algorithms and representation schemes—remains a major challenge across problem domains. Several recent studies [2, 3, 4, 5, 1] have exhibited specific examples of models apparently re-discovering interpretable (and in some cases familiar) solutions to tasks like curve detection, sequence copying and modular arithmetic. Are these models the exception or the rule? Under what conditions do neural network models discover familiar algorithmic solutions to algorithmic tasks?

In this paper, we focus specifically on the problem of learning modular addition, training networks to compute sums like $8 + 6 = 2 \pmod{12}$. Modular arithmetic can be implemented with a simple geometric solution, familiar to anyone who has learned to read a clock: every integer is represented as an angle, input angles are added together, and the resulting angle evaluated to obtain a modular sum (Figure 1, left). Nanda et al. [1] show that specific neural network architectures, when trained to perform modular addition, implement this *Clock* algorithm. In this work, we show that the *Clock* algorithm is only one part of a more complicated picture of algorithm learning in deep networks. In particular, networks structurally similar to the ones trained by Nanda et al. preferentially implement a qualitatively different approach to modular arithmetic, which we term the *Pizza* algorithm (Figure 1, right), and sometimes even more complex solutions. Models exhibit sharp *algorithmic phase transitions* [6] between the *Clock* and *Pizza* algorithms as their width and attention strength very, and often implement multiple, imperfect copies of the *Pizza* algorithm in parallel.

---

[*]Equal contribution.
[1]Code is available at https://github.com/fjzzq2002/pizza.

37th Conference on Neural Information Processing Systems (NeurIPS 2023).

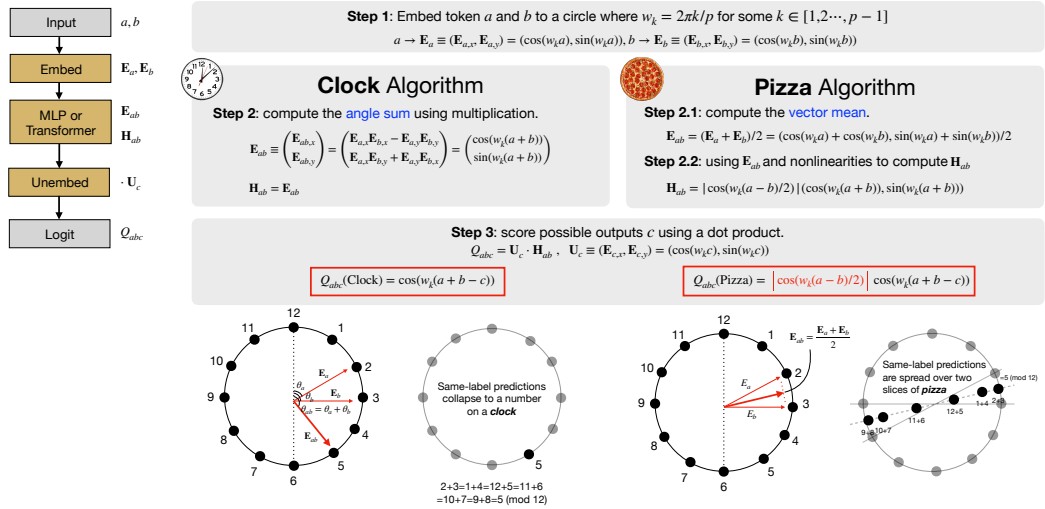

Figure 1: Illustration of the *Clock* and the *Pizza* Algorithm.

Our results highlight the complexity of mechanistic description in even models trained to perform simple tasks. They point to characterization of algorithmic phase spaces, not just single algorithmic solutions, as an important goal in algorithm-level interpretability.

**Organization** In Section 2, we review the *Clock* algorithm [1] and show empirical evidence of deviation from it in models trained to perform modular addition. In Section 3, we show that these deviations can be explained by an alternative *Pizza* algorithm. In Section 4, we define additional metrics to distinguish between these algorithms, and detect phase transitions between these algorithms (and others *Non-circular* algorithms) when architectures and hyperparameters are varied. We discuss the relationship between these findings and other work on model interpretation in Section 5, and conclude in Section 6.

## 2 Modular Arithmetic and the *Clock* Algorithm

**Setup** We train neural networks to perform modular addition $a + b = c \pmod{p}$, where $a, b, c = 0, 1, \cdots, p - 1$. We use $p = 59$ throughout the paper. In these networks, every integer $t$ has an associated embedding vector $\mathbf{E}_t \in \mathbb{R}^d$. Networks take as input embeddings $[\mathbf{E}_a, \mathbf{E}_b] \in \mathbb{R}^{2d}$ and predict a categorical output $c$. Both embeddings and network parameters are learned. In preliminary experiments, we train two different network architectures on the modular arithmetic task, which we refer to as: Model A and Model B. **Model A** is a one-layer ReLU transformer [7] with constant attention, while **Model B** is a standard one-layer ReLU transformer (see Appendix F.1 for details). As attention is not involved in Model A, it can also be understood as a ReLU MLP (Appendix G).

### 2.1 Review of the *Clock* Algorithm

As in past work, we find that after training both Model A and Model B, embeddings ($\mathbf{E}_a, \mathbf{E}_b$ in Figure 1) usually describe a circle [8] in the plane spanned by the first two principal components of the embedding matrix. Formally, $\mathbf{E}_a \approx [\cos(w_k a), \sin(w_k a)]$ where $w_k = 2\pi k/p$, $k$ is an integer in $[1, p - 1]$. Nanda et al. [1] discovered a circuit that uses these circular embeddings to implement an interpretable algorithm for modular arithmetic, which we call the *Clock* algorithm.

| Algorithm | Learned Embeddings | Gradient Symmetry | Required Non-linearity |
|---|---|---|---|
| Clock | Circle | No | Multiplication |
| Pizza | Circle | Yes | Absolute value |
| Non-circular | Line, Lissajous-like curves, etc. | N/A | N/A |

Table 1: Different neural algorithms for modular addition

"If a meeting starts at 10, and lasts for 3 hours, then it will end at 1." This familiar fact is a description of a modular sum, $10+3 = 1 \pmod{12}$, and the movement of a clock describes a simple algorithm for modular arithmetic: the numbers 1 through 12 are arranged on a circle in $360°/12 = 30°$ increments, angles of $10 \times 30°$ and $3 \times 30°$ are added together, then this angle is evaluated to determine that it corresponds to $1 \times 30°$.

Remarkably, Nanda et al. [1] find that neural networks like our Model B implement this *Clock* algorithm, visualized in Figure 1 (left): they represent tokens $a$ and $b$ as 2D vectors, and adding their polar angles using trigonometric identities. Concretely, the *Clock* algorithm consists of three steps: In step 1, tokens $a$ and $b$ are embedded as $\mathbf{E}_a = [\cos(w_k a), \sin(w_k a)]$ and $\mathbf{E}_b = [\cos(w_k b), \sin(w_k b)]$, respectively, where $w_k = 2\pi k/p$ (an everyday clock has $p = 12$ and $k = 1$). Then the polar angles of $\mathbf{E}_a$ and $\mathbf{E}_b$ are added (in step 2) and extracted (in step 3) via trigonometric identities. For each candidate output $c$, we denote the logit $Q_{abc}$; the predicted output is $c^* = \mathrm{argmax}_c Q_{abc}$.

Crucial to this algorithm is the fact that the attention mechanism can be leveraged to perform multiplication. What happens in model variants when the attention mechanism is absent, as in Model A? We find two pieces of evidence of deviation from the *Clock* algorithm in Model A.

## 2.2 First Evidence for *Clock* Violation: Gradient Symmetricity

Since the *Clock* algorithm has logits:

$$Q_{abc}^{\text{Clock}} = (\mathbf{E}_{a,x}\mathbf{E}_{b,x} - \mathbf{E}_{a,y}\mathbf{E}_{b,y})\mathbf{E}_{c,x} + (\mathbf{E}_{a,x}\mathbf{E}_{b,y} + \mathbf{E}_{a,y}\mathbf{E}_{b,x})\mathbf{E}_{c,y}, \qquad (1)$$

(see Figure 1) the gradients of $Q_{abc}$ generically lack permutation symmetry in argument order: $\nabla_{\mathbf{E}_a} Q_{abc} \neq \nabla_{\mathbf{E}_b} Q_{abc}$. Thus, if learned models exhibit permutation symmetry ($\nabla_{\mathbf{E}_a} Q_{abc} = \nabla_{\mathbf{E}_b} Q_{abc}$), they must be implementing some other algorithm.

We compute the 6 largest principal components of the input embedding vectors. We then compute the gradients of output logits (unnormalized log-probabilities from the model) with respect to the input embeddings. We then project them onto these 6 principal components (since the angles relevant to the *Clock* and *Pizza* algorithms are encoded in the first few principal components). These projections are shown in Figure 2. While Model B demonstrates asymmetry in general, Model A exhibits gradient symmetry.

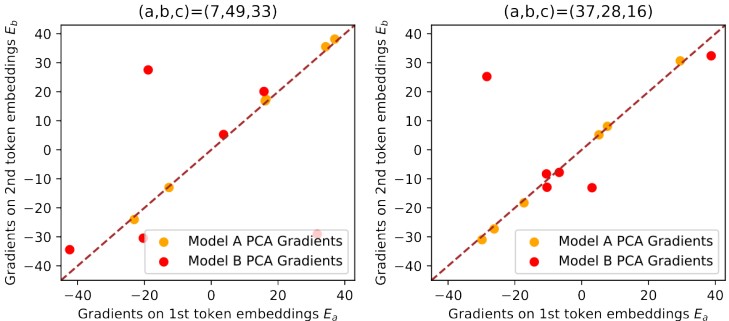

Figure 2: Gradients on first six principal components of input embeddings. $(a, b, c)$ in the title stands for taking gradients on the output logit $c$ for input $(a, b)$. x and y axes represent the gradients for embeddings of the first and the second token. The dashed line $y = x$ signals a symmetric gradient.

## 2.3 Second Evidence for *Clock* Violation: Logit Patterns

Inspecting models' outputs, in addition to inputs, reveals further differences. For each input pair $(a, b)$, we compute the output logit assigned to the correct label $a + b$. We visualize these *correct logits* from Models A and B in Figure 3. Notice that the rows are indexed by $a - b$ and the columns by $a + b$. From Figure 3, we can see that the correct logits of Model A have a clear dependency on $a - b$ in that within each row, the correct logits are roughly the same, while this pattern is not observed in Model B. This suggests that Models A and B are implementing different algorithms.

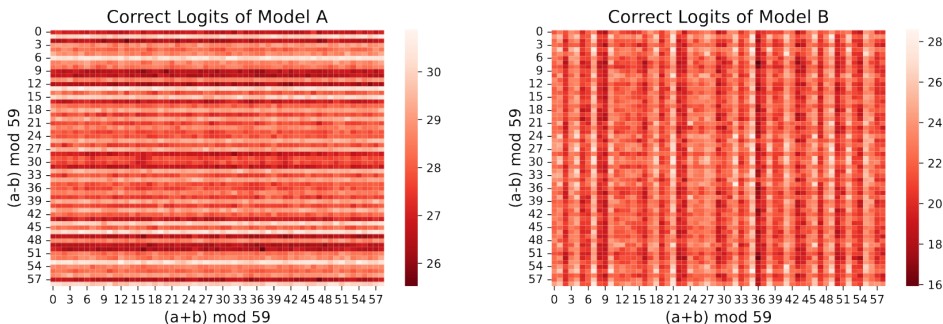

Figure 3: Correct Logits of Model A & Model B. The correct logits of Model A (left) have a clear dependence on $a - b$, while those of Model B (right) do not.

# 3 An Alternative Solution: the *Pizza* Algorithm

How does Model A perform modular arithmetic? Whatever solution it implements must exhibit gradient symmetricity in Figure 2 and the output patterns in Figure 3. In this section, we describe a new algorithm for modular arithmetic, which we call the *Pizza* algorithm, and then provide evidence that this is the procedure implemented by Model A.

## 3.1 The *Pizza* Algorithm

Unlike the *Clock* algorithm, the *Pizza* algorithm operates *inside* the circle formed by embeddings (just as pepperoni are spread all over a pizza), instead of operating on the circumference of the circle. The basic idea is illustrated in Figure 1: given a fixed label $c$, for *all* $(a, b)$ with $a + b = c \pmod{p}$, the points $\mathbf{E}_{ab} = (\mathbf{E}_a + \mathbf{E}_b)/2$ lie on a line though the origin of a 2D plane, and the points closer to this line than to the lines corresponding to any other $c$ form two out of $2p$ mirrored "pizza slices", as shown at the right of the figure. Thus, to perform modular arithmetic, a network can determine which slice pair the average of the two embedding vectors lies in. Concretely, the *Pizza* algorithm also consists of three steps. Step 1 is the same as in the *Clock* algorithm: the tokens $a$ and $b$ are embedded at $\mathbf{E}_a = (\cos(w_k a), \sin(w_k a))$ and $\mathbf{E}_b = (\cos(w_k b), \sin(w_k b))$, respectively. Step 2 and Step 3 are different from the *Clock* algorithm. In Step 2.1, $\mathbf{E}_a$ and $\mathbf{E}_b$ are averaged to produce an embedding $\mathbf{E}_{ab}$. In Step 2.2 and Step 3, the polar angle of $\mathbf{E}_{ab}$ is (implicitly) computed by computing the logit $Q_{abc}$ for any possible outputs $c$. While one possibility of doing so is to take the absolute value of the dot product of $\mathbf{E}_{ab}$ with $(\cos(w_k c/2), \sin(w_k c/2))$, it is not commonly observed in neural networks (and will result in a different logit pattern). Instead, Step 2.2 transforms $\mathbf{E}_{ab}$ into a vector encoding $|\cos(w_k(a - b)/2)|(\cos(w_k(a + b)), \sin(w_k(a + b)))$, which is then dotted with the output embedding $U_c = (\cos(w_k c), \sin(w_k c))$. Finally, the prediction is $c^* = \operatorname{argmax}_c Q_{abc}$. See Appendix A and Appendix L for a more detailed analysis of a neural circuit that computes $\mathbf{H}_{ab}$ in a real network.

The key difference between the two algorithms lies in what non-linear operations are required: *Clock* requires multiplication of inputs in Step 2, while *Pizza* requires only absolute value computation, which is easily implemented by the ReLU layers. If neural networks lack inductive biases toward implementing multiplication, they may be more likely to implement *Pizza* rather than *Clock*, as we will verify in Section 4.

## 3.2 First Evidence for *Pizza*: Logit Patterns

Both the *Clock* and *Pizza* algorithms compute logits $Q_{abc}$ in Step 3, but they have different forms, shown in Figure 1. Specifically, $Q_{abc}(Pizza)$ has an extra multiplicative factor $|\cos(w_k(a - b)/2)|$ compared to $Q_{abc}(Clock)$. As a result, given $c = a + b$, $Q_{abc}(Pizza)$ is dependent on $a - b$, but $Q_{abc}(Clock)$ is not. The intuition for the dependence is that a sample is more likely to be classified correctly if $\mathbf{E}_{ab}$ is longer. The norm of this vector depends on $a - b$. As we observe in Figure 3, the logits in Model A indeed exhibit a strong dependence on $a - b$.

### 3.3 Second Evidence for *Pizza*: Clearer Logit Patterns via Circle Isolation

To better understand the behavior of this algorithm, we replace the embedding matrix $\mathbf{E}$ with a series of rank-2 approximations: using only the first and second principal components, or only the third and fourth, etc. For each such matrix, embeddings lie in a a two-dimensional subspace. For both Model A and Model B, we find that embeddings form a circle in this subspace (Figure 4 and Figure 5, bottom). We call this procedure *circle isolation*. Even after this drastic modification to the trained models' parameters, both Model A and Model B continue to behave in interpretable ways: a subset of predictions remain highly accurate, with this subset determined by the periodicity of the $k$ of the isolated circle. As predicted by the *Pizza* and *Clock* algorithms described in Figure 1, Model A's accuracy drops to zero at specific values of $a - b$, while Model B's accuracy is invariant in $a - b$. Applying circle isolation to Model A on the two principal components (one circle) yields a model with $32.8\%$ overall accuracy, while retaining the first six principal components (three circles) yields an overall accuracy of $91.4\%$. See Appendix D for more discussion. By contrast, Model B achieves $100\%$ when embeddings are truncated to the first six principal components. Circle isolation thus reveals an *error correction* mechanism achieved via ensembling: when an algorithm (clock or pizza) exhibits systematic errors on subset of inputs, models can implement multiple algorithm variants in parallel to obtain more robust predictions.

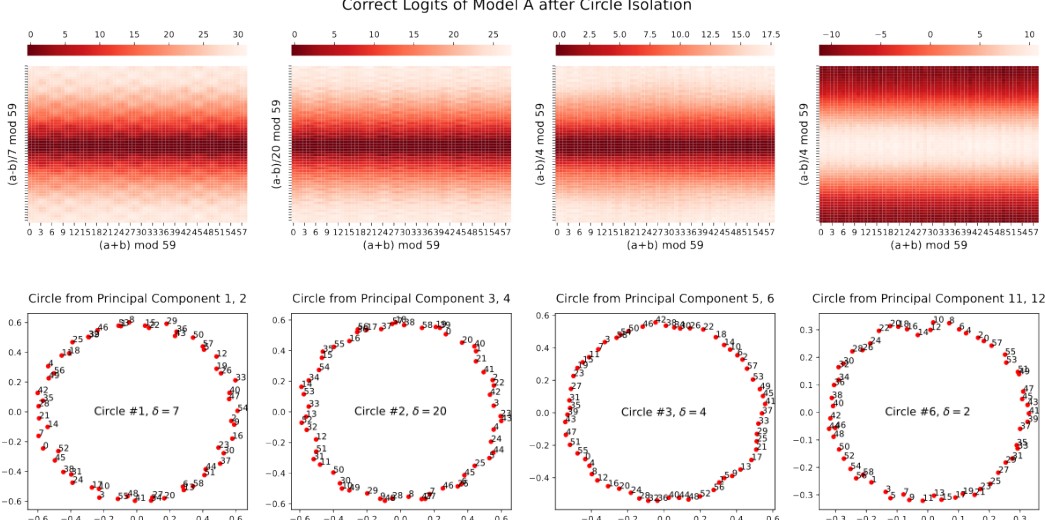

Figure 4: Correct logits of Model A (*Pizza*) after circle isolation. The rightmost pizza is accompanying the third pizza (discussed in Section 3.4 and Appendix D). *Top:* The logit pattern depends on $a - b$. *Bottom:* Embeddings for each circle.

Using these isolated embeddings, we may additionally calculate the isolated logits directly with formulas in Figure 1 and compare with the actual logits from Model A. Results are displayed in Table 2. We find that $Q_{abc}(Pizza)$ explains substantially more variance than $Q_{abc}(Clock)$.

**Why do we only analyze correct logits?** The logits from the *Pizza* algorithm are given by $Q_{abc}(Pizza) = |\cos(w_k(a - b)/2)| \cos(w_k(a + b - c))$. By contrast, the *Clock* algorithm has logits $Q_{abc}(Clock) = \cos(w_k(a + b - c))$. In a word, $Q_{abc}(Pizza)$ has an extra multiplicative factor $|\cos(w_k(a - b)/2)|$ compared to $Q_{abc}(Clock)$. By constraining $c = a + b$ (thus $\cos(w_k(a + b - c)) = 1$), the factor $|\cos(w_k(a - b)/2)|$ can be identified.

**(Unexpected) dependence of logits $Q_{abc}(Clock)$ on $a + b$:** Although our analysis above expects logits $Q_{abc}(Clock)$ not to depend on $a - b$, they do not predict its dependence on $a + b$. In Figure 5, we surprisingly find that $Q_{abc}(Clock)$ is sensitive to this sum. Our conjecture is that Step 1 and Step 2 of the *Clock* are implemented (almost) noiselessly, such that same-label samples collapse to the same point after Step 2. However, Step 3 (classification) is imperfect after circle isolation, resulting in fluctuations of logits. Inputs with common sums $a + b$ produce the same logits.

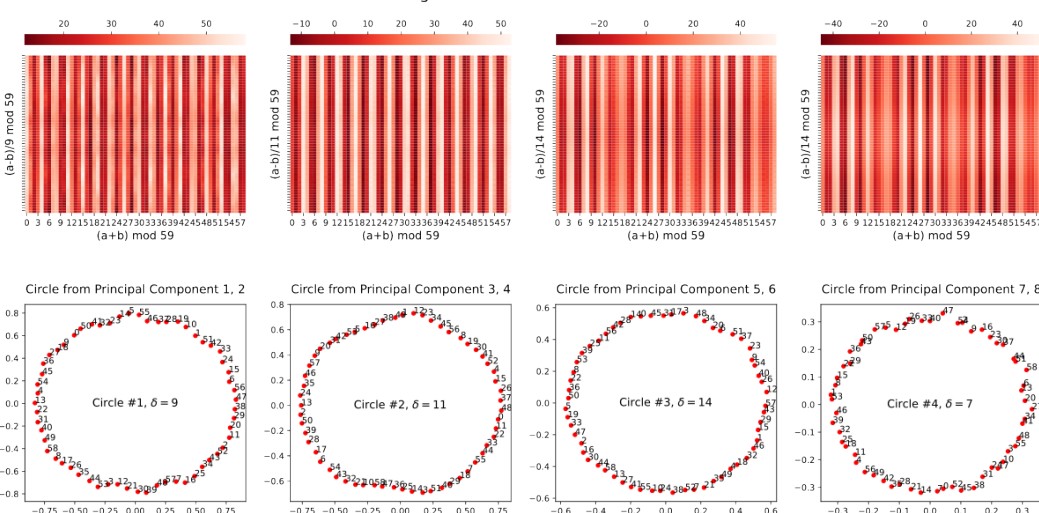

Figure 5: Correct logits of Model B (*Clock*) after circle isolation. *Top:* The logit pattern depends on $a + b$. *Bottom:* Embeddings for each circle.

| Circle | $w_k$ | $Q_{abc}$(clock) FVE | $Q_{abc}$(pizza) FVE |
|--------|-------|---------------------|---------------------|
| #1 | $2\pi/59 \cdot 17$ | 75.41% | 99.18% |
| #2 | $2\pi/59 \cdot 3$ | 75.62% | 99.18% |
| #3 | $2\pi/59 \cdot 44$ | 75.38% | 99.28% |

Table 2: After isolating circles in the input embedding, fraction of variance explained (FVE) of **all** Model A's output logits ($59 \times 59 \times 59$ of them) by various formulas. Both model output logits and formula results' are normalized to mean 0 variance 1 before taking FVE. $w_k$'s are calculated according to the visualization. For example, distance between 0 and 1 in Circle #1 is 17, so $w_k = 2\pi/59 \cdot 17$.

## 3.4 Third Evidence for *Pizza*: Accompanied & Accompanying Pizza

The Achilles' heel of the *Pizza* algorithm is antipodal pairs. If two inputs $(a, b)$ happen to lie antipodally, then their middle point will lie at the origin, where the correct "pizza slice" is difficult to identify. For example in Figure 1 right, antipodal pairs are (1,7), (2,8), (3,9) etc., whose middle points all collapse to the origin, but their class labels are different. Models cannot distinguish between, and thus correctly classify, these pairs. Even for odd $p$'s where there are no strict antipodal pairs, approximately antipodal pairs are also more likely to be classified incorrectly than non-antipodal pairs.

Intriguingly, neural networks find a clever way to compensate for this failure mode. we find that pizzas usually come with "accompanying pizzas". An accompanied pizza and its accompanying pizza complement each other in the sense that near-antipodal pairs in the accompanied pizza become adjacent or close (i.e, very non-antipodal) in the accompanying pizza. If we denote the difference between adjacent numbers on the circle as $\delta$ and $\delta_1$, $\delta_2$ for accompanied and accompanying pizzas, respectively, then $\delta_1 = 2\delta_2 \pmod{p}$. In the experiment, we found that pizzas #1/#2/#3 in Figure 4 all have accompanying pizzas, which we call pizzas #4/#5/#6 (see Appendix D for details). However, these accompanying pizzas do not play a significant role in final model predictions [2]. We conjecture that training dynamics are as follows: (1) At initialization, pizzas #1/#2/#3 correspond to three different "lottery tickets" [9]. (2) In early stages of training, to compensate the weaknesses (antipodal pairs) of pizzas #1/#2/#3, pizzas #4/#5/#6 are formed. (3) As training goes on (in the presence of weight decay), the neural network gets pruned. As a result, pizzas #4/#5/#6 are not significantly involved in prediction, although they continue to be visible in the embedding space.

---

[2] Accompanied pizzas #1/#2/#3 can achieve 99.7% accuracy, but accompanying pizzas #4/#5/#6 can only achieve 16.7% accuracy.

# 4 The Algorithmic Phase Space

In Section 3, we have demonstrated a typical *Clock* (Model A) and a typical *Pizza* (Model B). In this section, we study how architectures and hyperparametes govern the selection of these two algorithmic "phases". In Section 4.1, we propose quantitative metrics that can distinguish between *Pizza* and *Clock*. In Section 4.2, we observe how these metrics behave with different architectures and hyperparameters, demonstrating sharp phase transitions. The results in this section focus *Clock* and *Pizza* models, but other algorithmic solutions to modular addition are also discovered, and explored in more detail in Appendix B.

## 4.1 Metrics

We wish to study the distribution of *Pizza* and *Clock* algorithms statistically, which will require us to distinguish between two algorithms automatically. In order to do so, we formalize our observations in Section 2.2 and 2.3, arriving at two metrics: **gradient symmetricity** and **distance irrelevance**.

### 4.1.1 Gradient Symmetricity

To measure the symmetricity of the gradients, we select some input-output group $(a, b, c)$, compute the gradient vectors for the output logit at position $c$ with respect to the input embeddings, and then compute the cosine similarity. Taking the average over many pairs yields the gradient symmetricity.

**Definition 4.1** (Gradient Symmetricity). *For a fixed set $S \subseteq \mathbb{Z}_p^3$ of input-output pairs[3], define* **gradient-symmetricity** *of a network $M$ with embedding layer $E$ as*

$$s_g \equiv \frac{1}{|S|} \sum_{(a,b,c) \in S} sim \left( \frac{\partial Q_{abc}}{\partial \mathbf{E}_a}, \frac{\partial Q_{abc}}{\partial \mathbf{E}_b} \right),$$

*where $sim(a, b) = \frac{a \cdot b}{|a||b|}$ is the cosine-similarity, $Q_{abc}$ is the logit for class $c$ given input $a$ and $b$. It is clear that $s_g \in [-1, 1]$.*

As we discussed in Section 2.2, the *Pizza* algorithm has symmetric gradients while the *Clock* algorithm has asymmetric ones. Model A and Model B in Section 3 have gradient symmetricity 99.37% and 33.36%, respectively (Figure 2).

### 4.1.2 Distance Irrelevance

To measure the dependence of correct logits on differences between two inputs, which reflect the distances of the inputs on the circles, we measure how much of the variance in the correct logit matrix depends on it. We do so by comparing the average standard deviation of correct logits from inputs with the same differences and the standard deviation from all inputs.

**Definition 4.2** (Distance Irrelevance). *For some network $M$ with correct logit matrix $L$ ($L_{i,j} = Q_{ij,i+j}$), define its* **distance irrelevance** *as*

$$q \equiv \frac{\frac{1}{p} \sum_{d \in \mathbb{Z}_p} \text{std} \left( L_{i,i+d} \mid i \in \mathbb{Z}_p \right)}{\text{std} \left( L_{i,j} \mid i, j \in \mathbb{Z}_p^2 \right)},$$

*where $\text{std}$ computes the standard deviation of a set. It is clear that $q \in [0, 1]$.*

Model A and Model B in Section 3 give distance irrelevance 0.17 and 0.85, respectively (Figure 3). A typical distance irrelevance from the *Pizza* algorithm ranges from 0 to 0.4 while a typical distance irrelevance from *Clock* algorithm ranges from 0.4 to 1.

### 4.1.3 Which Metric is More Decisive?

When the two metrics have conflicting results, which one is more decisive? We consider distance irrelevance as the decisive factor of the *Pizza* algorithm, as the output logits being dependent on the distance is highly suggestive of *Pizza*. On the other hand, gradient symmetricity can be used to rule out the *Clock* algorithm, as it requires multiplying (transformed) inputs which will result in asymmetric gradients. Figure 6 confirmed that at low distance irrelevance (suggesting pizza) the gradient symmetricity is almost always close to 1 (suggesting non-clock).

---

[3]To speed-up the calculations, in our experiments $S$ is taken as a random subset of $\mathbb{Z}_p^3$ of size 100.

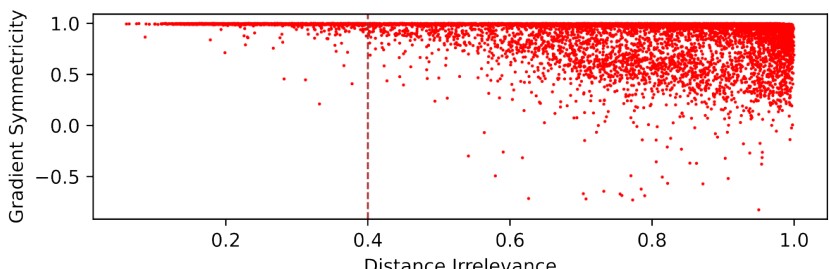

Figure 6: Distance irrelevance vs gradient symmetricity over all the main experiments.

## 4.2 Identifying algorithmic phase transitions

How do models "choose" whether to implement the *Clock* or *Pizza* algorithm? We investigate this question by interpolating between Model A (transformer without attention) and Model B (transformer with attention). To do so, we introduce a new hyperparameter $\alpha$ we call the **attention rate**.

For a model with attention rate $\alpha$, we modify the attention matrix $M$ for each attention head to be $M' = M\alpha + J(1 - \alpha)$. In other words, we modify this matrix to consist of a linear interpolation between the all-one matrix and the original attention (post-softmax), with the rate $\alpha$ controlling how much of the attention is kept. The transformer with and without attention corresponds to the case where $\alpha = 1$ (attention kept) and $\alpha = 0$ (constant attention matrix). With this parameter, we can control the balance of attention versus linear layers in transformers.

We performed the following set of experiments on transformers (see Appendix F.1 for architecture and training details). (1) One-layer transformers with width 128 and attention rate uniformly sampled in $[0, 1]$ (Figure 7). (2) One-layer transformers with width log-uniformly sampled in $[32, 512]$ and attention rate uniformly sampled in $[0, 1]$ (Figure 7). (3) Transformers with 2 to 4 layers, width 128 and attention rate uniformly sampled in $[0, 1]$ (Figure 11).

**The *Pizza* and the *Clock* algorithms are the dominating algorithms with circular embeddings.** For circular models, most observed models either have low gradient symmetricity (corresponding to the *Clock* algorithm) or low distance irrelevance (corresponding to the *Pizza* algorithm).

**Two-dimensional phase change observed for attention rate and layer width.** For the fixed-width experiment, we observed a clear phase transition from the *Pizza* algorithm to the *Clock* algorithm (characterized by gradient symmetricity and distance irrelevance). We also observe an almost linear phase boundary with regards to both attention rate and layer width. In other words, the attention rate transition point increases as the model gets wider.

**Dominance of linear layers determines whether the *Pizza* or the *Clock* algorithm is preferred.** For one-layer transformers, we study the transition point against the attention rate and the width:

- The *Clock* algorithm dominates when the attention rate is higher than the phase change point, and the *Pizza* algorithm dominates when the attention rate is lower than the point. Our explanation is: At a high attention rate, the attention mechanism is more prominent in the network, giving rise to the clock algorithm. At a low attention rate, the linear layers are more prominent, giving rise to the pizza algorithm.

- The phase change point gets higher when the model width increases. Our explanation is: When the model gets wider, the linear layers become more capable while the attention mechanism receive less benefit (attentions remain scalars while outputs from linear layers become wider vectors). The linear layer therefore gets more prominence with a wider model.

**Possibly hybrid algorithms between the *Clock* and the *Pizza* algorithms.** The continuous phase change suggests the existence of networks that lie between the *Clock* and the *Pizza* algorithms. This is achievable by having some principal components acting as the *Clock* and some principal components acting as the *Pizza*.

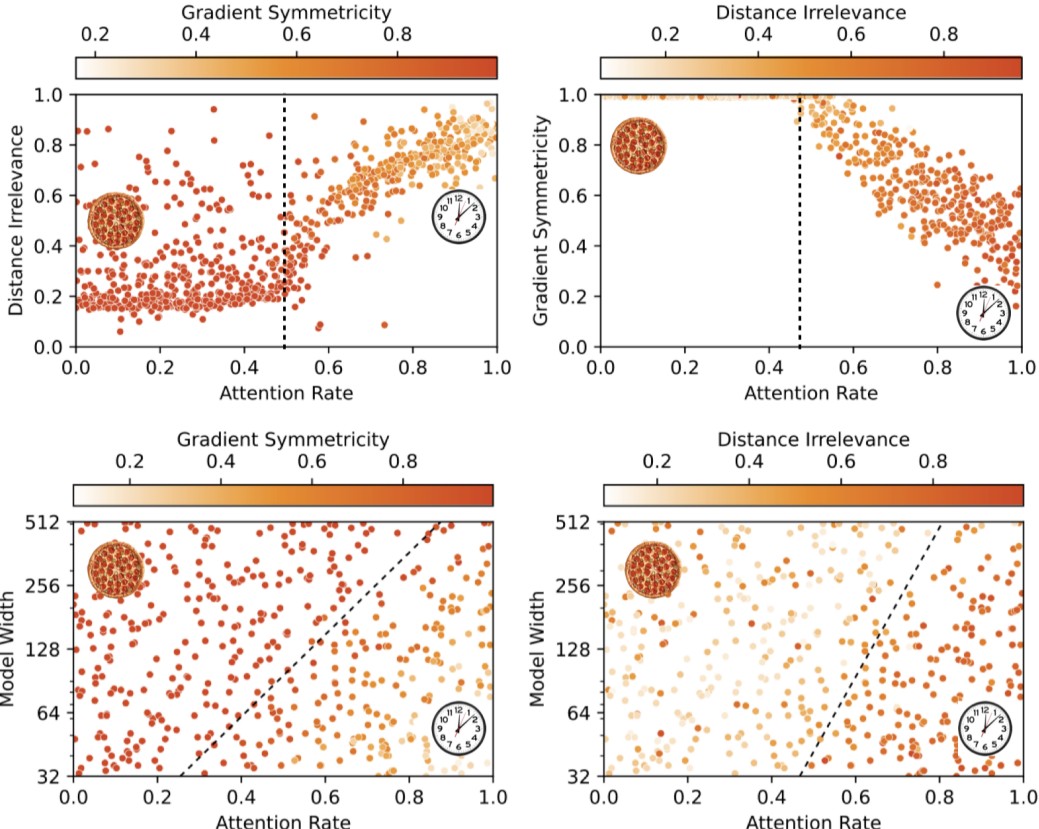

Figure 7: Training results from 1-layer transformers. Each point in the plots represents a training run reaching circular embeddings and 100% validation accuracy. See Appendix C for additional plots. *Top:* Model width fixed to be 128. *Bottom:* Model width varies. The phase transition lines are calculated by logistic regression (classify the runs by whether gradient symmetricity > 98% and whether distance irrelevance < 0.6).

**Existence of non-circular algorithms.** Although our presentation focuses on circular algorithms (i.e., whose embeddings are circular), we find non-circular algorithms (i.e., whose embeddings do not form a circle when projected onto any plane) to be present in neural networks. See Appendix B for preliminary findings. We find that deeper networks are more likely to form non-circular algorithms. We also observe the appearance of non-circular networks at low attention rates. Nevertheless, the *Pizza* algorithm continues to be observed (low distance irrelevance, high gradient symmetricity).

## 5 Related Work

**Mechanistic interpretability** aims to mechanically understand neural networks by reverse engineering them [2, 3, 5, 4, 10, 11, 1, 12, 13, 14]. One can either look for patterns in weights and activations by studying single-neuron behavior (superposition [11], monosemantic neurons [15]), or study meaningful modules or circuits grouped by neurons [4, 14]. Mechanistic interpretability is closely related to training dynamics [8, 13, 1].

**Learning mathematical tasks**: Mathematical tasks provide useful benchmarks for neural network interpretability, since the tasks themselves are well understood. The setup could be learning from images [16, 17], with trainable embeddings [18], or with number as inputs [19, 5]. Beyond arithmetic relations, machine learning has been applied to learn other mathematical structures, including geometry [20], knot theory [21] and group theory [22].

**Algorithmic phase transitions**: Phase transitions are present in classical algorithms [23] and in deep learning [6, 24, 25]. Usually the phase transition means that the algorithmic performance sharply

changes when a parameter is varied (e.g., amount of data, network capacity etc). However, the phase transition studied in this paper is *representational*: both clock and pizza give perfect accuracy, but arrive at answers via different interal computations. These model-internal phase transitions are harder to study, but closer to corresponding phenomena in physical systems [24].

**Algorithm learning in neural networks**: Emergent abilities in deep neural networks, especially large language models, have recently attracted significant attention [26]. An ability is "emergent" if the performance on a subtask suddenly increases with growing model sizes, though such claims depend on the choice of metric [27]. It has been hypothesized that the emergence of specific capability in a model corresponds to the emergence of a modular circuit responsible for that capability, and that emergence of some model behaviors thus results from a sequence of quantized circuit discovery steps [5].

## 6 Conclusions

We have offered a closer look at recent findings that familiar algorithms arise in neural networks trained on specific algorithmic tasks. In modular arithmetic, we have shown that such algorithmic discoveries are not inevitable: in addition to the *Clock* algorithm reverse-engineered by [1], we find other algorithms (including a *Pizza* algorithm, and more complicated procedures) to be prevalent in trained models. These different algorithmic phases can be distinguished using a variety of new and existing interpretability techniques, including logit visualization, isolation of principle components in embedding space, and gradient-based measures of model symmetry. These techniques make it possible to *automatically* classify trained networks according to the algorithms they implement, and reveal algorithmic phase transitions in the space of model hyperparameters. Here we found specifically that the emergence of a *Pizza* or *Clock* algorithm depends on the relative strength of linear layers and attention outputs. We additionally showed that these algorithms are not implemented in isolation; instead, networks sometimes ensemble multiple copies of an algorithm in parallel. These results offer exciting new challenges for mechanistic interpretability: (1) How to find, classify, and interpret unfamiliar algorithms in a systematic way? (2) How to disentangle multiple, parallel algorithm implementations in the presence of ensembling?

**Limitations** We have focused on a single learning problem: modular addition. Even in this restricted domain, qualitatively different model behaviors emerge across architectures and seeds. Significant additional work is needed to scale these techniques to the even more complex models used in real-world tasks.

**Broader Impact** We believe interpretability techniques can play a crucial role in creating and improving safe AI systems. However, they may also be used to build more accurate systems, with the attendant risks inherent in all dual-use technologies. It is therefore necessary to exercise caution and responsible decision-making when deploying such techniques.

## Acknowledgement

We would like to thank Mingyang Deng and anonymous reviewers for valuable and fruitful discussions and MIT SuperCloud for providing computation resources. ZL and MT are supported by the Foundational Questions Institute, the Rothberg Family Fund for Cognitive Science and IAIFI through NSF grant PHY-2019786. JA is supported by a gift from the OpenPhilanthropy Foundation.

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

# Supplementary material

## A   Mathematical Analysis and An Example of Pizza Algorithm

In the pizza algorithm, we have $E_{ab} = \cos(w_k(a-b)/2) \cdot (\cos(w_k(a+b)/2), \sin(w_k(a+b)/2))$, as $\cos x + \cos y = \cos((x-y)/2)(2\cos((x+y)/2))$ and $\sin x + \sin y = \cos((x-y)/2)(2\sin((x+y)/2))$.

To get $|\cos(w_k(a-b)/2)|(\cos(w_k(a+b)), \sin(w_k(a+b)))$, we generalize this to $|\cos(w_k(a-b)/2)|\cos(w_k(a+b-u))$ (the two given cases correspond to $u=0$ and $u=\pi/2/w_k$).

$$|(\cos(w_k u/2), \sin(w_k u/2)) \cdot E_{ab}| = |\cos(w_k(a-b)/2)\cos(w_k(a+b-u)/2)|$$
$$|(-\sin(w_k u/2), \cos(w_k u/2)) \cdot E_{ab}| = |\cos(w_k(a-b)/2)\sin(w_k(a+b-u)/2)|$$

thus their difference will be equal to

$$|\cos(w_k(a-b)/2)|(|\cos(w_k(a+b-u)/2)| - |\sin(w_k(a+b-u)/2)|).$$

Now notice $|\cos(t)| - |\sin(t)| \approx \cos(2t)$ for any $t \in \mathbb{R}$ (Figure 8), so the difference is approximately $|\cos(w_k(a-b)/2)|\cos(w_k(a+b-u))$.

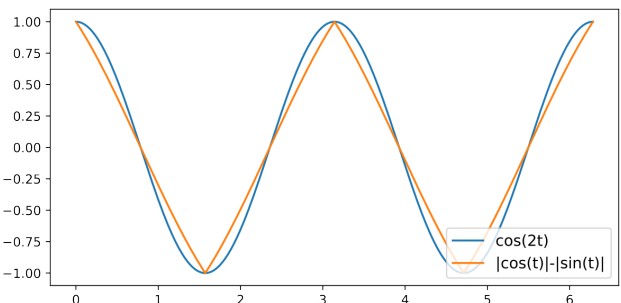

Figure 8: $|\cos(t)| - |\sin(t)|$ is approximately $\cos(2t)$ for any $t \in \mathbb{R}$

Plugging in $u=0$ and $u=\pi/2/w_k$ as mentioned, we get the following particular implementation of the pizza algorithm.

**Algorithm: Pizza, Example**

Step 1   On given input $a$ and $b$, circularly embed them to two vectors on the circumference $(\cos(w_k a), \sin(w_k a))$ and $(\cos(w_k b), \sin(w_k b))$.

Step 2   Compute:

$$\alpha = |\cos(w_k a) + \cos(w_k b)|/2 - |\sin(w_k a) + \sin(w_k b)|/2$$
$$\approx |\cos(w_k(a-b)/2)|\cos(w_k(a+b))$$
$$\beta = |\cos(w_k a) + \cos(w_k b) + \sin(w_k a) + \sin(w_k b)|/(2\sqrt{2})$$
$$- |\cos(w_k a) + \cos(w_k b) - \sin(w_k a) - \sin(w_k b)|/(2\sqrt{2})$$
$$= |\cos(w_k a - \pi/4) + \cos(w_k b - \pi/4)|/2 - |\sin(w_k a - \pi/4) + \sin(w_k b - \pi/4)|/2$$
$$\approx |\cos(w_k(a-b)/2)|\cos(w_k(a+b) - \pi/2) = |\cos(w_k(a-b)/2)|\sin(w_k(a+b))$$

Step 3   Output of this pizza is computed as a dot product.

$$Q'_{abc} = \alpha\cos(w_k c) + \beta\sin(w_k c) \approx |\cos(w_k(a-b)/2)|\cos(w_k(a+b-c))$$

Similar circuits are observed in the wild, but instead of the above two-term approximation, a more complicated one is observed. See Appendix L for details.

The extra $|\cos(w_k(a-b)/2)|$ term is not a coincidence. We can generalize our derivation as the following.

**Lemma A.1.** *A symmetric function $f(x, y)$ that is a linear combination of $\cos x, \sin x, \cos y, \sin y$[4] can always be written as $\cos((x - y)/2)g(x + y)$ for some function $g$.*

*Proof.* Notice $\cos x + \cos y = \cos((x - y)/2)(2\cos((x + y)/2))$ and $\sin x + \sin y = \cos((x - y)/2)(2\sin((x + y)/2))$, so $\alpha(\cos x + \cos y) + \beta(\sin x + \sin y) = \cos((x - y)/2)(2\alpha\cos((x + y)/2) + 2\beta\sin((x + y)/2))$. $\square$

This is why we consider the output pattern with the $|\cos(w_k(a - b)/2)|$ terms rather than the actual computation circuits as the determinant feature of the pizza algorithm.

## B    Non-Circular algorithms

One thing that further complicates our experiment is the existence of non-circular embeddings. While only circular algorithms are reported in the previous works [8, 1], many non-circular embeddings are found in our experiments, e.g., 1D lines or 3D Lissajous-like curves, as shown in Figure 9. We leave the detailed analysis of these non-circular algorithms for future study. Since circular algorithms are our primary focus of study, we propose the following metric **circularity** to filter out non-circular algorithms. The metric reaches maximum 1 when the principal components aligns with cosine waves.

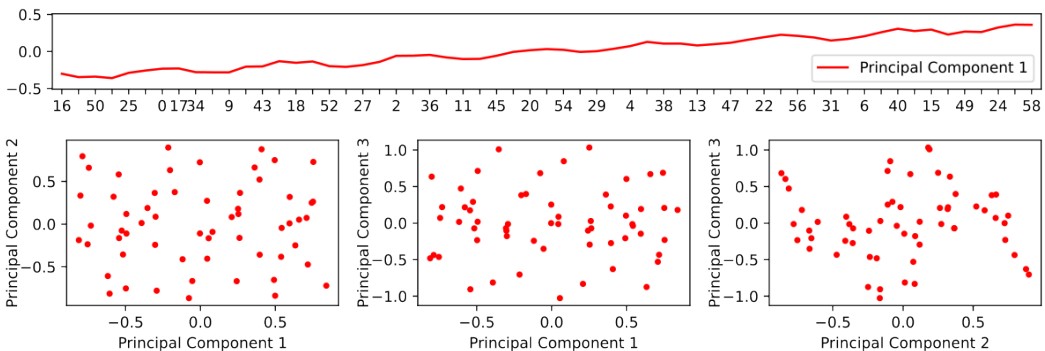

Figure 9: Visualization of the principal components of input embeddings for two trained non-circular models. *Top:* A line-like first principal component. Notice the re-arranged x axis (token id). *Bottom:* First three principal components forming a three-dimensional non-circular pattern. Each point represents the embedding of a token.

**Definition B.1** (Circularity). *For some network, suppose the $l$-th principal component of its input embeddings is $v_{l,0}, v_{l,1}, \cdots, v_{l,p-1}$, define its **circularity** based on first four components as*

$$c = \frac{1}{4}\sum_{l=1}^{4}\left(\max_{k\in[1,2,\cdots,p-1]}\left(\frac{2}{p\sum_{j=0}^{p-1}v_{l,j}^2}\left|\sum_{j=0}^{p-1}v_{l,j}e^{2\pi i \cdot jk/p}\right|^2\right)\right)$$

*where $i$ is the imaginary unit. $c \in [0, 1]$ by Fourier analysis. $c = 1$ means first four components are Fourier waves.*

Both Model A and Model B in Section 3 have a circularity around $99.8\%$ and we consider models with circularity $\geq 99.5\%$ **circular**.

## C    More Results from the Main Experiments

Here we provide Figure 7 with non-circular networks unfiltered (Figure 10). We can see more noise emerging in the plot. We also provide the training results from multi-layer transformers (Figure 11).

---

[4]The actual neural networks could be more complicated - even if our neural network is locally linear and symmetric, locally they could be asymmetric (e.g. $|x| + |y|$ could locally be $x - y$). Nevertheless, the pattern is observed in our trained networks.

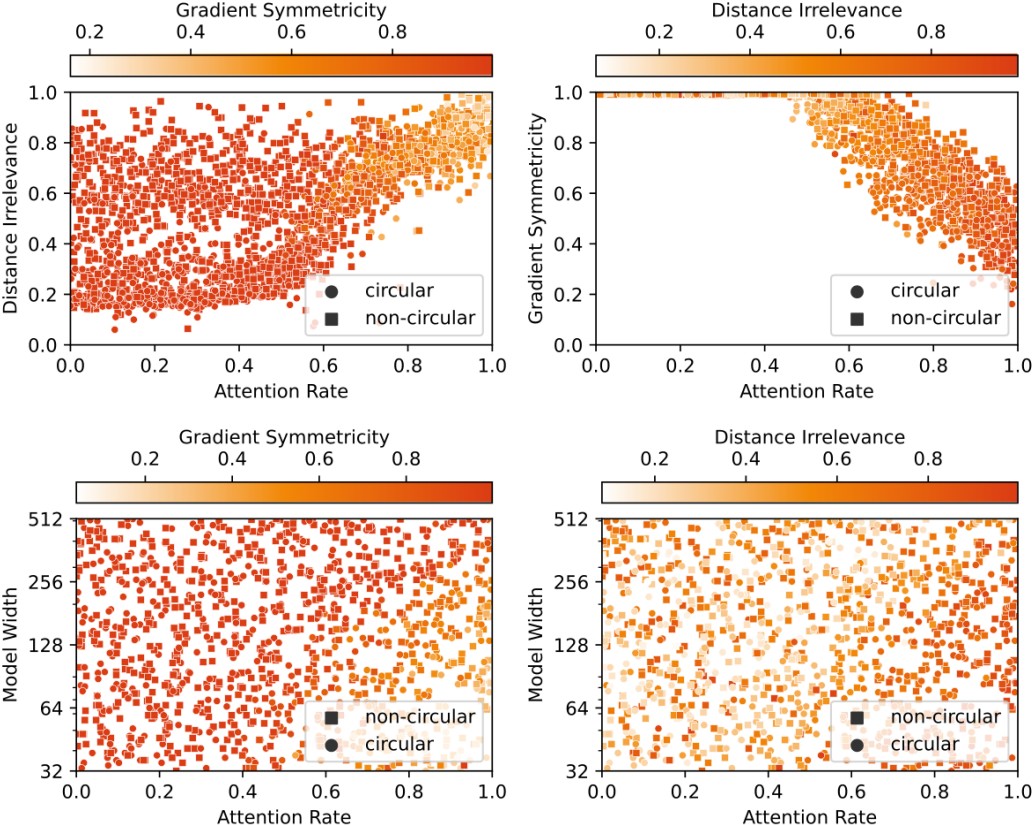

Figure 10: Training results from 1-layer transformers. Each point in the plots represents a training run reaching 100% validation accuracy. Among all the trained 1-layer transformers, 34.31% are circular. *Top:* Model width fixed to be 128. *Bottom:* Model width varies.

## D    Pizzas Come in Pairs

Cautious readers might notice that the pizza algorithm is imperfect - for near antipodal points, the sum vector will have a very small norm and the result will be noise-sensitive. While the problem is partially elevated by the use of multiple circles instead of one, we also noticed another pattern emerged: **accompanying pizzas**.

The idea is the following: suppose the difference between adjacent points is $2k \bmod p$, then the antipodal points have difference $\pm k$. Therefore, if we arrange a new circle with a difference $k$ for adjacent points, we will get a pizza that works **best** for formerly antipodal points.

**Algorithm: Accompanying Pizza**

Step 1   Take $w_k$ as of the accompanied pizza. On given input $a$ and $b$, circularly embed them to two vectors on the circumference $(\cos(2w_k a), \sin(2w_k a))$ and $(\cos(2w_k b), \sin(2w_k b))$.

Step 2   Compute the midpoint:

$$s = \frac{1}{2}(\cos(2w_k a) + \cos(2w_k b), \sin(2w_k a) + \sin(2w_k b))$$

Step 3   Output of this pizza is computed as a dot product.

$$A_c = -(\cos(w_k c), \sin(w_k c)) \cdot s$$

This is exactly what we observed in Model A (Table 3, Figure 13). With the six circles (pizzas and accompanying pizzas) included in the embedding, Model A also gets 100% accuracy.

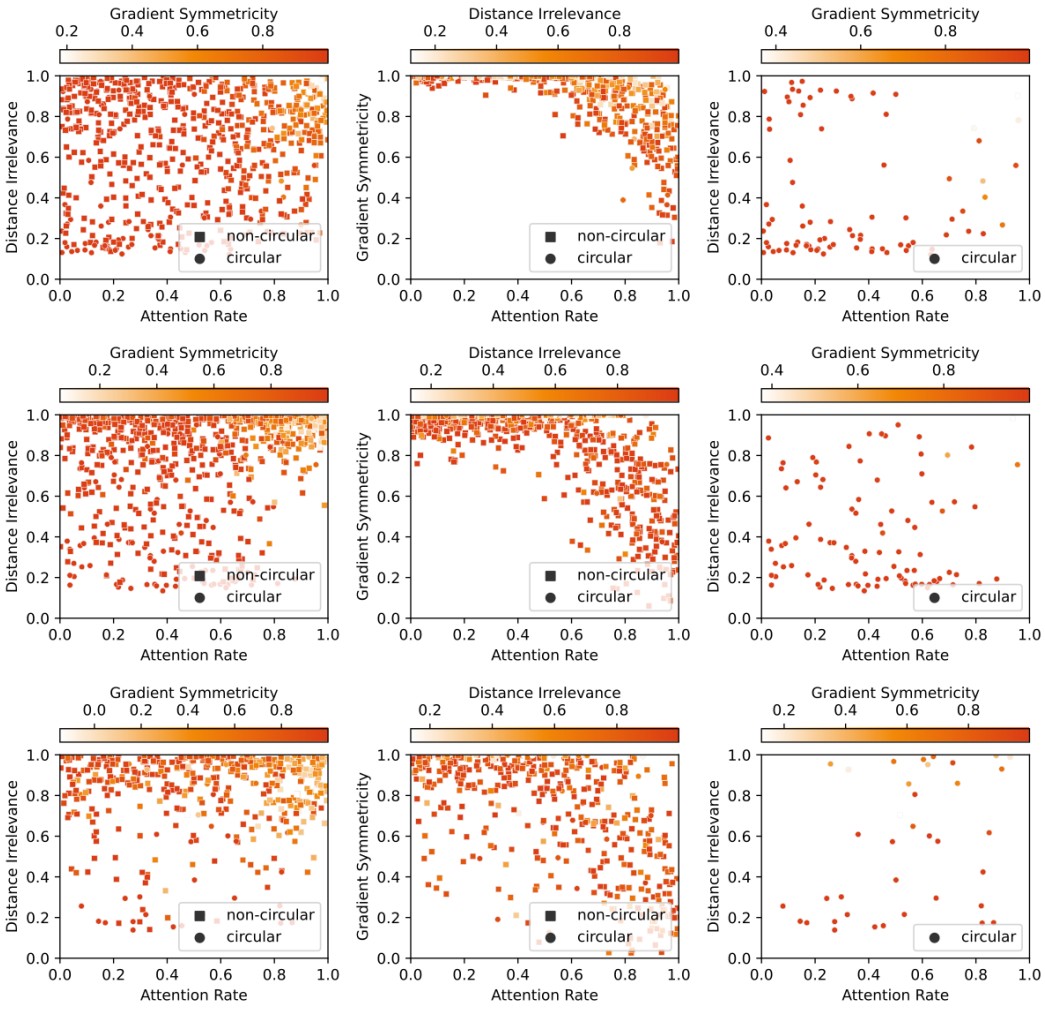

Figure 11: Training results from transformers with 2, 3 and 4 layers. Among all the trained transformers with 2, 3 and 4 layers, 9.95%, 11.55% and 6.08% are circular, respectively.

Figure 12: An Illustration on the Accompanying Pizza Algorithm

| Circle | $w_k$ | $A_c$ FVE |
|---|---|---|
| #4 (accompanying #1) | $2\pi/59 \cdot 17$ | 97.56% |
| #5 (accompanying #2) | $2\pi/59 \cdot 3$ | 97.23% |
| #6 (accompanying #3) | $2\pi/59 \cdot 44$ | 97.69% |

Table 3: After isolating accompanying circles in the input embedding, fraction of variance explained (FVE) of **all** Model A's output logits by various formulas. Both model output logits and formula results' are normalized to mean 0 variance 1 before taking FVE. Accompanying and accompanied pizza have the same $w_k$.

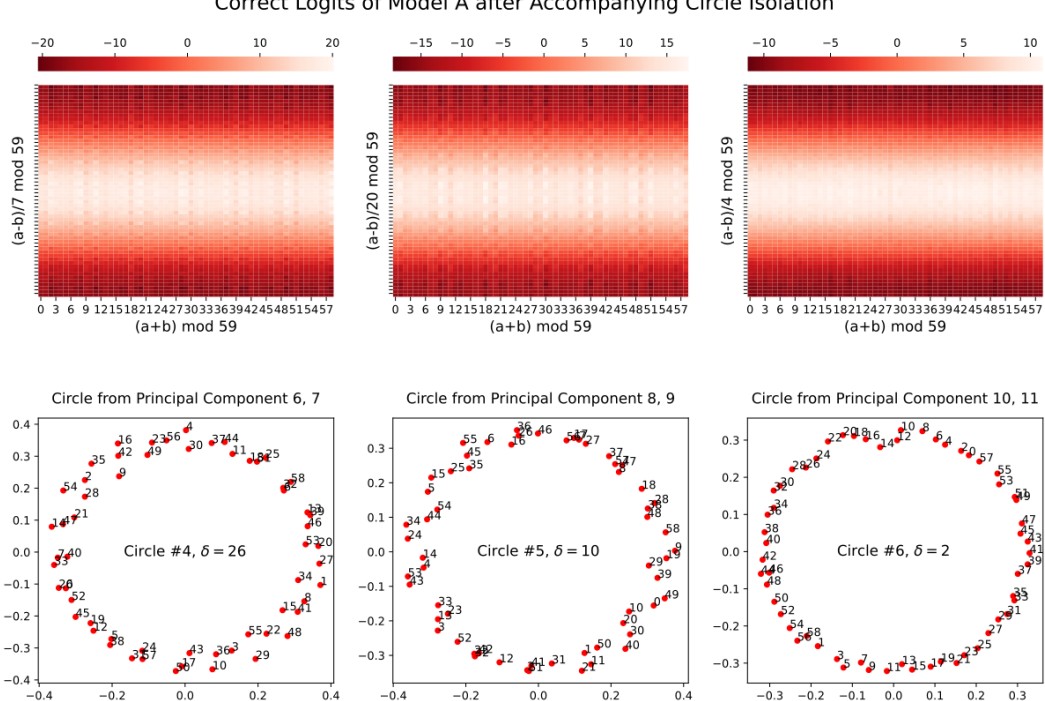

Figure 13: Correct logits of Model A (*Pizza*) after circle isolation. Only accompanying pizzas are displayed. Notice the complementing logit pattern (Figure 4).

## E   Results in Other Linear Architectures

While this is not the primary focus of our paper, we also ran experiments on the following four different linear model setups (see Section F.2 for setup details).

- For all the models, we first encode input tokens $(a, b)$ with a trainable embedding layer $W_E$: $x_1 = W_{E,a}$, $x_2 = W_{E,b}$ (positional embedding removed for simplicity). $L_1, L_2, L_3$ are trainable linear layers. The outmost layers (commonly referred as unembed layers) have no biases and the other layers have biases included for generality.

- Model $\alpha$: calculate output logits as $L_2(\text{ReLU}(L_1(x_1 + x_2)))$.

- Model $\beta$: calculate output logits as $L_3(\text{ReLU}(L_2(\text{ReLU}(L_1(x_1 + x_2)))))$.

- Model $\gamma$: calculate output logits as $L_3(\text{ReLU}(L_2(\text{ReLU}(L_1(x_1) + L_1(x_2)))))$.

- Model $\delta$: calculate output logits as $L_2(\text{ReLU}(L_1([x_1; x_2])))$

  ($[x_1; x_2]$ stands for the concatenation of $x_1$ and $x_2$)

The results are shown in Figure 14. Rather surprisingly, Model $\alpha$, Model $\beta$ and Model $\delta$ gave radically different results. Model $\beta$ and Model $\gamma$ are very similar, and in general they are more pizza-like

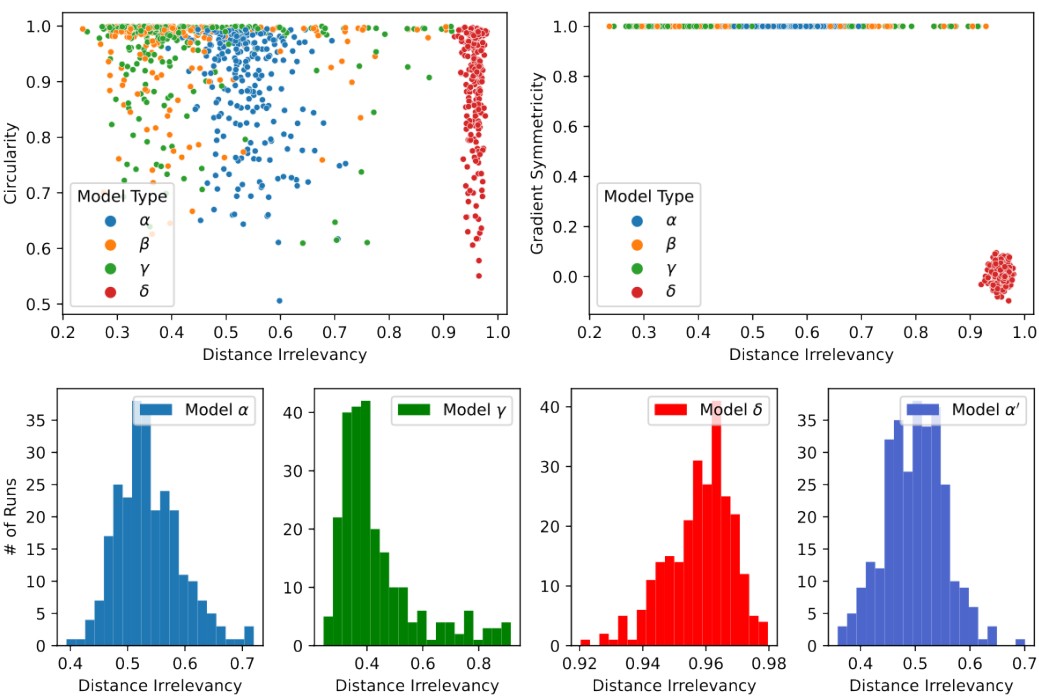

Figure 14: Training results from linear models. Each point in the first-row plots represents a training run. The second row are histograms for distance irrelevancy of each model type.

than Model $\alpha$, with lower distance irrelevancy and higher circularity. This could be explained by the addition of an extra linear layer.

However, Model $\delta$ gave very different results from Model $\alpha$ although they are both one-layer linear models. It is more likely to be non-circular and have very high distance irrelevancy in general. In other words, concatenating instead of adding embeddings yields radically different behaviors in one-layer linear model. This result, again, alarmed us the significance of induction biases in neural networks.

We also want to note that using different embeddings on two tokens of Model $\alpha$ doesn't resolve the discrepancy. The following model

- Model $\alpha'$: calculate output logits as $L_2(\text{ReLU}(L_1(x_1 + x_2)))$ where $x_1 = W_{E,a}^A$, $x_2 = W_{E,b}^B$ on input $(a, b)$ and $W_E^A, W_E^B$ are different embedding layers.

gives roughly the same result as of Model $\alpha$ (Figure 14, lower right corner).

Figure 15 shows the correct logits after circle isolation (Section 3.3) of a circular model from Model $\beta$ implementing the pizza algorithm. Figure 16 shows the correct logits after circle isolation (Section 3.3) of a circular model from Model $\delta$. We can see the pattern is similar but different from the one of clock algorithm (Figure 5). We leave the study of such models to future work.

# F    Architecture and Training Details

## F.1    Transformers

Here we describe our setup for the main experiments. See Appendix E and Appendix I for experiments on different setups.

**Architecture**    We train bidirectional transformers (attention unmasked) to perform modular addition $\mod p$ where $p = 59$. To calculate $(a + b) \mod p$, the input is provided to the model as a sequence

Figure 15: Correct logits from Model $\beta$ after circle isolation.

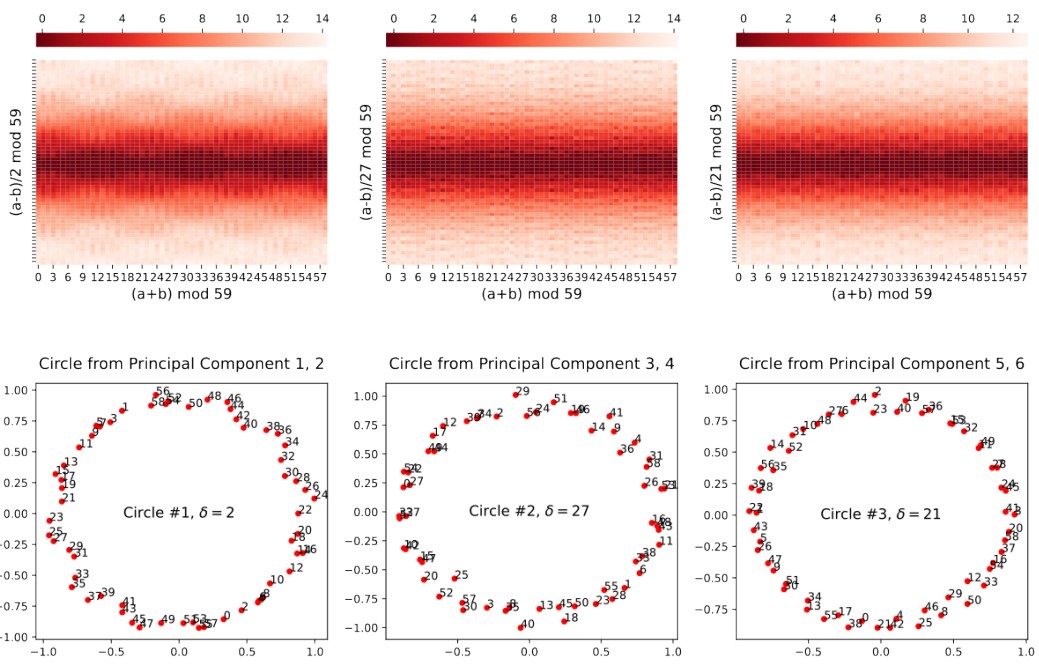

Figure 16: Correct logits from Model $\delta$ after circle isolation.

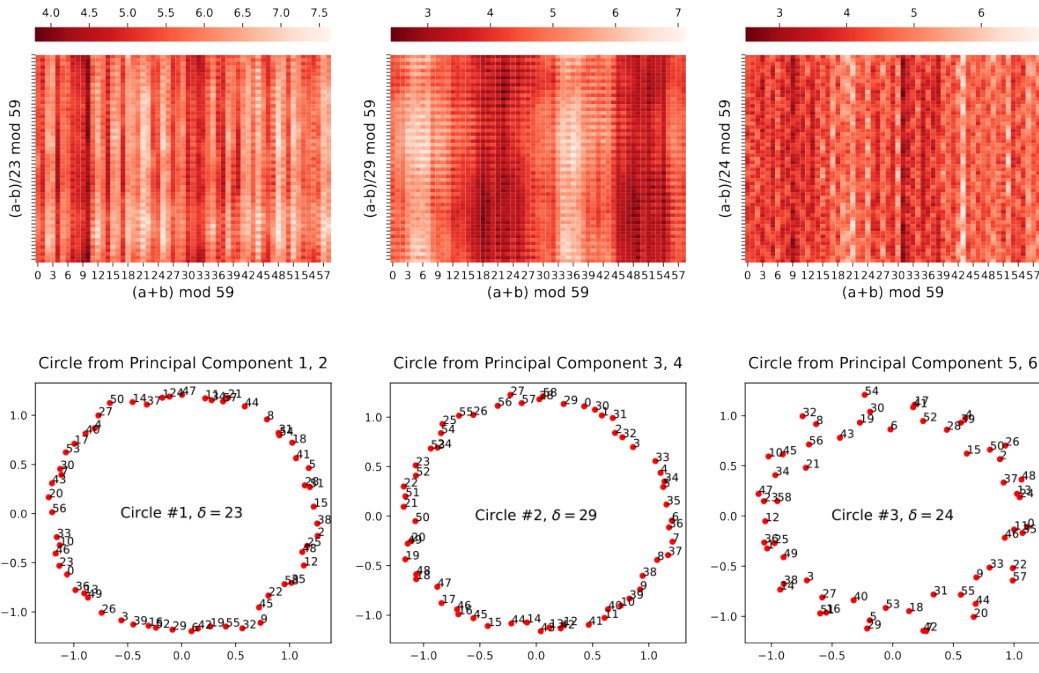

of two tokens $[a, b]$. The output logit at the last token is considered as the output of the model. For a transformer with "width" $d$, the input embedding and the residue stream will be $d$-dimensional, 4 attention heads of $\lfloor d/4 \rfloor$ dimensions will be employed, and the MLP will be of $4d$ hidden units. By default $d = 128$ is chosen. ReLU is used as the activation function and layer normalization isn't applied. The post-softmax attention matrix is interpolated between an all-one matrix and original as

specified by the attention rate (Section 4.2). We want to point out that the setup of constant-attention transformers is also considered in the previous work [28].

**Data**    Among all possible data points ($p^2 = 3481$ of them), we randomly select $80\%$ as training samples and $20\%$ as validation samples. This choice (small $p$ and high training data fraction) helps accelerating the training.

**Training**    We used AdamW optimizer [29] with learning rate $\gamma = 0.001$ and weight decay factor $\beta = 2$. We do not use minibatches and the shuffled training data is provided as a whole batch in every epoch. For each run, we start the training from scratch and train for $20,000$ epoches. We removed the runs that did not reach $100\%$ validation accuracy at the end of the training (majority of the runs reached $100\%$).

## F.2    Linear Models

Here we describe our setup for the linear model experiments (Appendix E).

**Architecture**    We train several types of linear models to perform modular addition $\mod p$ where $p = 59$. The input embedding, residue stream and hidden layer are all $d = 256$ dimensional. ReLU is used as the activation function. The actual structures of network types are specified in Appendix E.

**Data & Training**    Same as in the previous section (Section F.1).

## F.3    Computing Resources

A total of 226 GPU days of NVidia V100 is spent on this project, although we expect a replication would take significantly fewer resources.

# G    Mathematical Description of Constant-Attention transformer

In this section, we examine the structure of constant-attention transformers loosely following the notation of [10].

Denote the weight of embedding layer as $W_E$, the weight of positional embedding as $W_{\text{pos}}$, the weight of the value and output matrix of the $j$-th head of the $t$-th layer as $W_V^{t,j}$ and $W_O^{t,j}$, the weights and biases of the input linear map of MLP in the $t$-th layer as $W_{\text{in}}^t$ and $b_{\text{in}}^t$, the corresponding weights and biases of the output linear map as $W_{\text{out}}^t$ and $b_{\text{out}}^t$, and the weight of the unembedding layer as $W_U$. Notice that the query and the key matrices are irrelevant as the attention matrix is replaced with an all-one matrix. Denote $x^j$ as the value of residue stream vector after the first $j$ layers and denote $c_i$ as the character in the $i$-th position. We use subscripts like $x_t$ to denote taking a specific element of vector.

We can formalize the logit calculation as the following:

- Embedding: $x_i^0 = W_{E,c_i} + W_{\text{pos},i}$.
- For each layer $t$ from 1 to $n_{\text{layer}}$:
    - **Constant** Attention: $w_i^t = x_i^{t-1} + \sum_j W_O^{t,j} W_V^{t,j} \sum_k x_k^{t-1}$.
    - MLP: $x^t = w^t + b_{\text{out}}^t + W_{\text{out}}^t \text{ReLU}(b_{\text{in}}^t + W_{\text{in}}^t w^t)$.
- Output: $O = W_U x^{n_{\text{layer}}}$.

In the particular case where the input length is 2, the number of layer is 1, and we focus on the logit of the last position, we may restate as the following (denote $z$ as $x^1$ and $y$ as $w^1$):

- Embedding: $x_1 = W_{E,c_1} + W_{\text{pos},1}, \ x_2 = W_{E,c_2} + W_{\text{pos},2}$.
- Constant Attention: $y = x_2 + \sum_j W_O^j W_V^j (x_1 + x_2)$.
- MLP: $z = y + b_{\text{out}}^t + W_{\text{out}}^t \text{ReLU}(b_{\text{in}}^t + W_{\text{in}}^t y)$.

- Output: $o = W_U z$.

If we remove the skip connections, the network after embedding could be seen as

$$o = L_U \left( L_{\text{out}} \left( \text{ReLU} \left( L_{\text{in}} \left( \sum_j L_O^j \left( L_V^j \left( x_1 + x_2 \right) \right) \right) \right) \right) \right)$$

where $L_V^j, L_O^j, L_{\text{in}}, L_{\text{out}}, L_U$ are a series of linear layers corresponding to the matrices.

## H  Pizza with Attention

Extrapolating from Figure 7, we trained transformers with width $1024$ and attention rate 1 (normal attention). After several tries, we are able to observe a trained circular model with distance irrelevance $0.156$ and gradient symmetricity $0.995$, which fits our definition of *Pizza* (Figure 17).

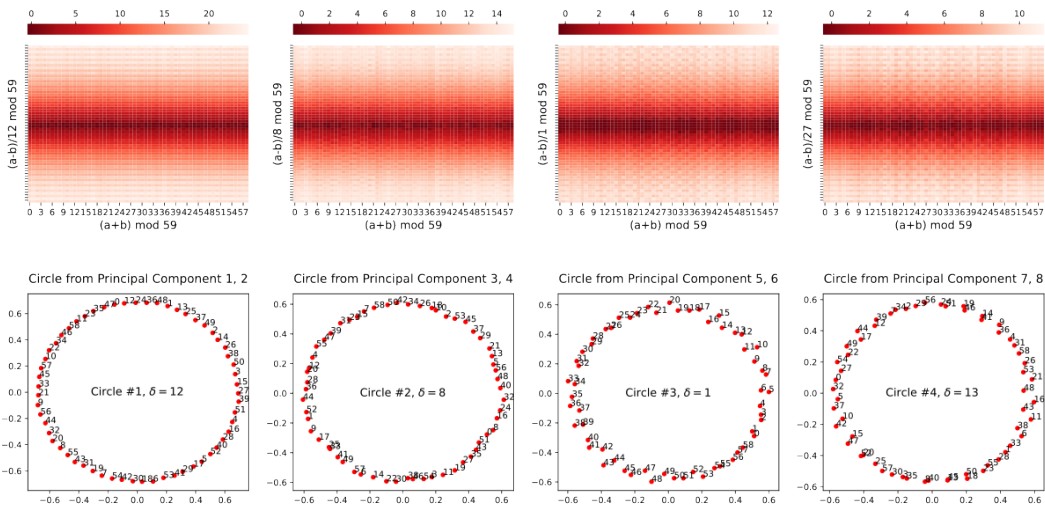

Figure 17: Correct logits of the trained model in Section H after circle isolation (Section 3.3).

## I  Results on Slightly Different Setups

We considered the following variations of our setups (Appendix F.1, Section 4), for which the existence of pizzas and clocks as well as the phase changes are still observed.

**GeLU instead of ReLU**  We conducted the same 1-layer transformer experiment with activation function GeLU instead of ReLU. Very similar results are observed (Figure 18).

**Encode Two Tokens Differently**  We conducted the 1-layer transformer experiments but with different embedding for the two tokens. Again very similar results are observed (Figure 19). We also discovered that the two tokens' embeddings are often aligned to implement the *Pizza* and *Clock* algorithm (Figure 20).

**Adding Equal Sign**  We conducted the 1-layer transformer experiment with an equal sign added. Very similar results are observed (Figure 21).

## J  Pizza Occurs Early in the Clock Training

We plotted intermediate states during the training of a model with attention (attention rate 1). Pizza-like pattern was observed early in the training, but the pattern gradually disappeared during the run (Figure 22).

Figure 18: Training results from 1-layer transformers with GeLU instead of ReLU as the activation function. Each point in the plots represents a training run that reached 100% validation accuracy.

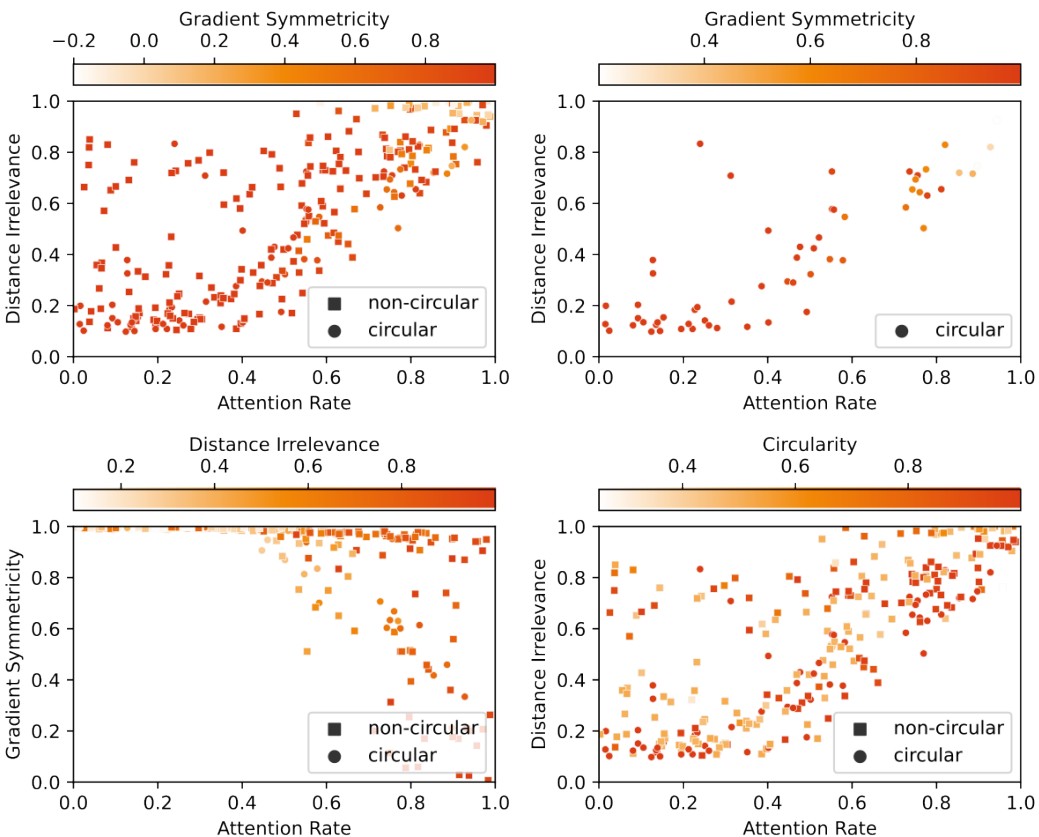

Figure 19: Training results from 1-layer transformers where the two tokens use different embeddings (feed $[a, b + p]$ to the model on input $(a, b)$; $2p$ tokens are handled in the embedding layer). Each point in the plots represents a training run that reached 100% validation accuracy. We did not use circularity to filter the result because it is no longer well-defined.

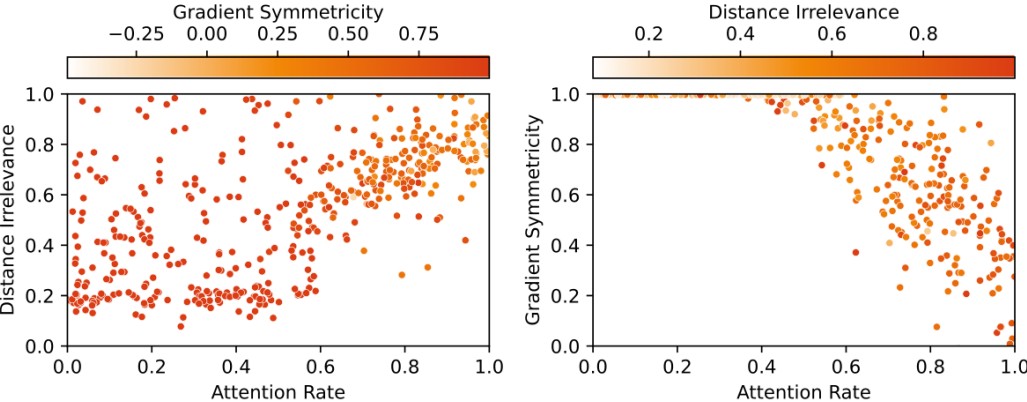

Figure 20: Correct logits after circle isolation from a trained model where two tokens use different embeddings. The blue points represent the embeddings for the first token and the green points represent the embeddings for the second token. The model is implementing the *Pizza* algorithm. The correct logit pattern is shifted comparing to the previous patterns because the embeddings of two tokens do not line up exactly. For example, the third circle has near-maximum correct logit for $a = 6, b = 3$ (the two points lining up on the top) and $(a - b)/18 \equiv 10 \pmod{59}$. This is the reason that the correct logit pattern appears to be shifted 10 units down.

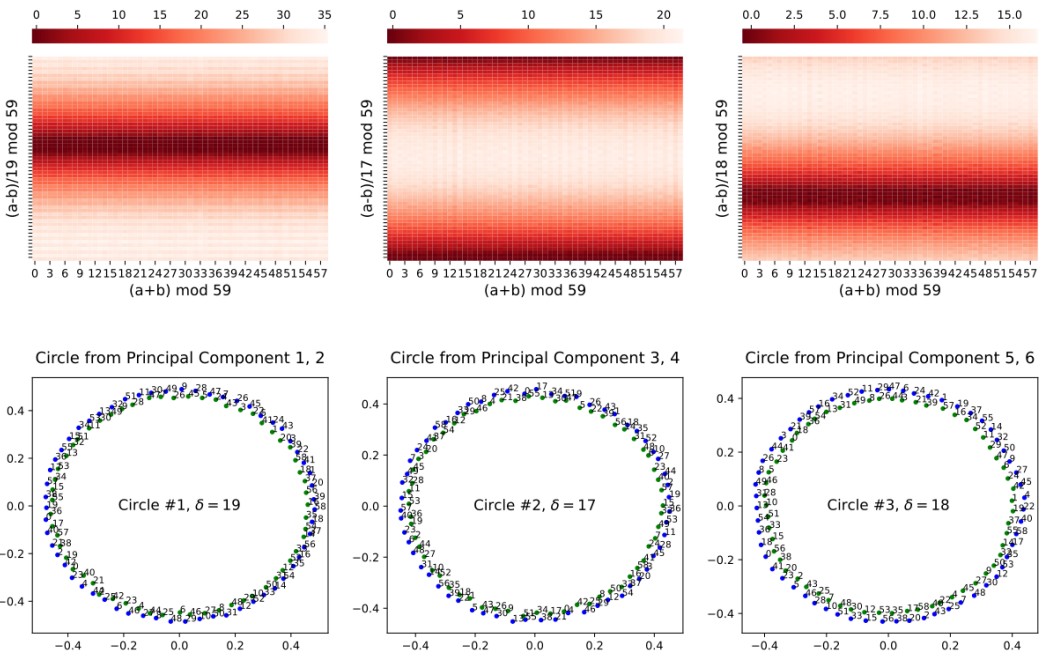

Figure 21: Training results from 1-layer transformers where an equal sign is added (feed $[a, b, =]$ to the model on input $(a, b)$ where = is a special token; $p + 1$ tokens are handled in the embedding layer; context length of the model becomes 3). Each point in the plots represents a training run that reached 100% validation accuracy. We did not use circularity to filter the result because it is no longer well-defined.

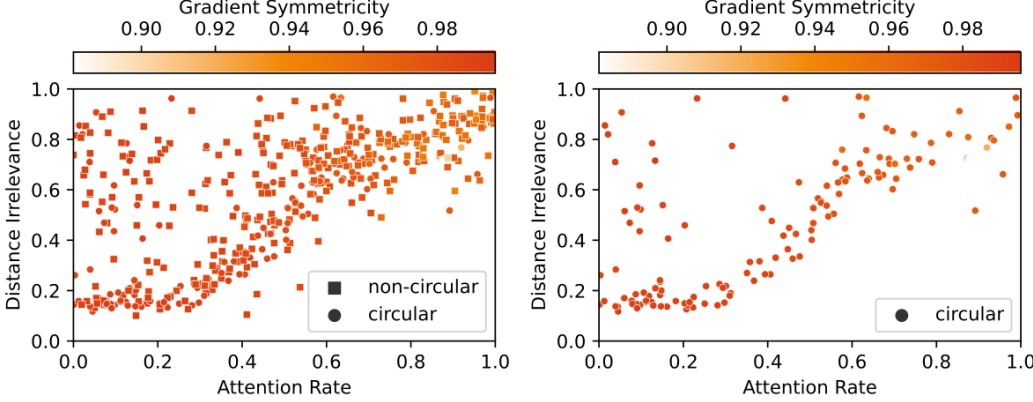

## K  Accompanying Pizza Occurs Early in the Pizza Training

We plotted intermediate states during the training of a model without attention (attention rate 0). We observed the early emergence of a pattern similar to accompanying pizza in training runs (Figure

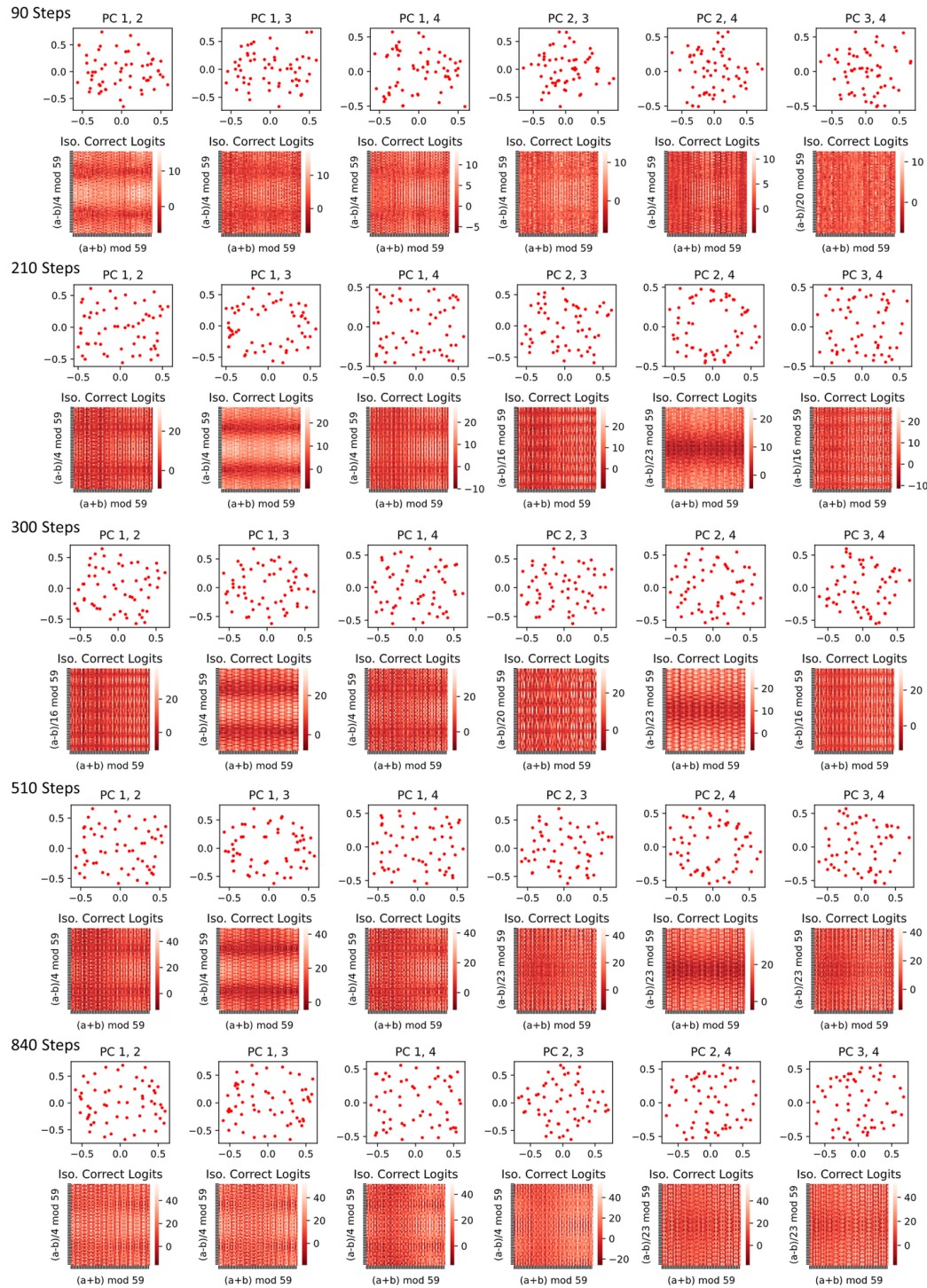

Figure 22: For a 1-layer transformer with attention, correct logits after principal component (possibly non-circle) isolations at various states during the training. The pizza-like pattern gradually *desolved*.

23) and removing that circle brings accuracy down from 99.7% to 97.9%. They are less helpful later in the network (removing accompanying pizzas in trained Model A only brings accuracy down to 99.7%).

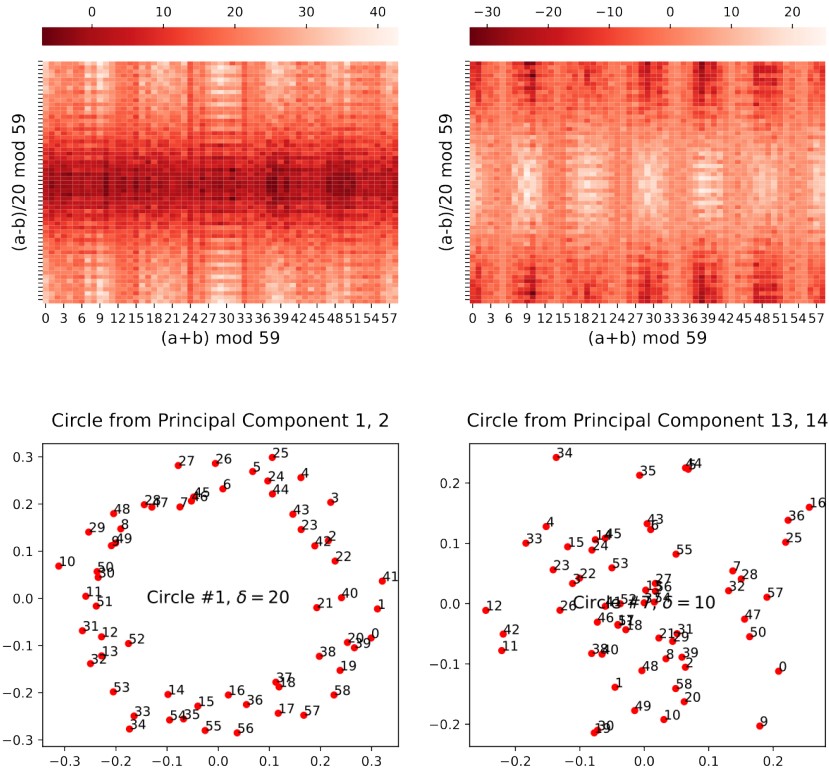

Figure 23: Immediate state after 600 epochs of training for a 1-layer transformer with constant attention.

## L  A Closer Look at a Linear Pizza Model

In this section, we provide a full picture of the linear model shown in Figure 15 by investigating the actual weights in the model.

### L.1  Model Structure

As described in Appendix E, on input $(a, b)$, the output logits of the model is computed as

$$L_3(\text{ReLU}(L_2(\text{ReLU}(L_1(\text{Embed}[a] + \text{Embed}[b]))))).$$

Denote the weight of embedding layer as $W_E$, the weight of the unembedding layer ($L_3$) as $W_U$, and the weights and biases of $L_1$ and $L_2$ as $W_1, b_1$ and $W_2, b_2$, respectively, then the output logits on input $(a, b)$ can be written as

$$W_U \text{ReLU}(b_2 + W_2 \text{ReLU}(b_1 + W_1(W_E[a] + W_E[b]))).$$

### L.2  General Picture

We first perform principal component visualizations on the embedding and unembedding matrices. From Figure 24, we can see that the embedding and unembedding matrices formed *matching circles* (circles with the same gap $\delta$ between adjacent entries).

We now give the general overview of the circuit. Each pair of matching circles forms an instance of *Pizza* and they operate independently (with rather limited interference). Specifically for each pair,

- The embedding matrix first places the inputs $a, b$ on the circumference: $W'_E[a] \approx (\cos(w_k a), \sin(w_k a))$ and $W'_E[b] \approx (\cos(w_k b), \sin(w_k b))$ ($w_k = 2\pi k/p$ for some integer $k \in [1, p-1]$ as in Section 2.1; $W'_E$ stands for the two currently considered principal components of $W_E$; rotation and scaling omitted for brevity).

- The embeddings are added to get

$$(\cos(w_k a) + \cos(w_k b), \sin(w_k a) + \sin(w_k b))$$
$$= \cos(w_k(a-b)/2) \cdot (\cos(w_k(a+b)/2), \sin(w_k(a+b)/2))$$

- It is then passed through the first linear layer $L_1$. Each result entry pre-ReLU will thus be a linear combination of the two dimensions of the aforementioned vectors, i.e. $\cos(w_k(a-b)/2) \cdot (\alpha \cos(w_k(a+b)/2) + \beta \sin(w_k(a+b)/2))$ for some $\alpha, \beta$, which will become $|\cos(w_k(a-b)/2)||\alpha \cos(w_k(a+b)/2) + \beta \sin(w_k(a+b)/2))|$ after ReLU.

- These values are then passed through the second linear layer $L_2$. Empirically the ReLU is not observed to be effective as the majority of values is positive. The output entries are then simply linear combinations of aforementioned outputs of $L_1$.

- The unembedding matrix is finally applied. In the principal components we are considering, $W_U'[c] \approx (\cos(w_k c), \sin(w_k c))$. ($W_U'$ stands for the two currently considered principal components of $W_U$; rotation and scaling omitted for brevity) and these two principal components correspond to a linear combination of the output entries of $L_2$, which then correspond to a linear combination of the outputs of $L_1$ (thanks to the non-functional ReLU).

- Similar to the formula $|\sin(t)| - |\cos(t)| \approx \cos(2t)$ discussed in Appendix A, these linear combinations provide good approximations for $|\cos(w_k(a-b)/2)| \cos(w_k(a+b))$ and $|\cos(w_k(a-b)/2)| \sin(w_k(a+b))$. Finally we arrive at

$$|\cos(w_k(a-b)/2)|(\cos(w_k c)\cos(w_k(a+b)) + \sin(w_k c)\sin(w_k(a+b)))$$
$$= |\cos(w_k(a-b)/2)| \cos(w_k(a+b-c))$$

.

Figure 24: Visualization of the principal components of the embeddings and unembedding matrices.

### L.3  Aligning Weight Matrices

We first verify that the ReLU from the second layer is not functional. After removing it, the accuracy of the model remains $100\%$ and the cross-entropy loss actually *decreased* from $6.20 \times 10^{-7}$ to $5.89 \times 10^{-7}$.

Therefore, the model output can be approximately written as

$$W_U(b_2 + W_2 \text{ReLU}(b_1 + W_1(W_E[a] + W_E[b]))) = W_U b_2 + W_U W_2 \text{ReLU}(b_1 + W_1(W_E[a] + W_E[b])).$$

We now "align" the weight matrices $W_1$ and $W_2$ by mapping through the directions of the principal components of the embeddings and unembeddings. That is, we calculate how these matrices act on and onto the principal directions (consider $W_1 v$ for every principal direction $v$ in $W_E$ and $v^T W_2$ for every principal direction $v$ in $W_U$). We call the other dimension of aligned $W_1$ and $W_2$ output and source dimensions, respectively (Figure 25).

In the aligned weight matrices, we can see a clear domino-like pattern: in most output or source dimensions, only two principal components have significant non-zero values, and they correspond to a pair of matching circle, or a pizza. In this way, every immediate dimension serves for exactly one pizza, so the pizzas do not interfere with each other.

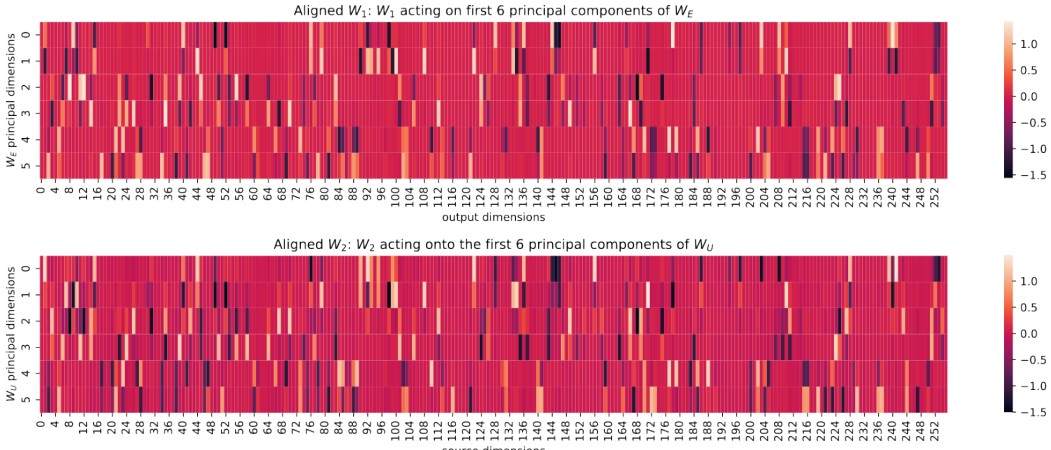

Figure 25: Visualization of the aligned $W_1$ and $W_2$.

## L.4   Approximation

Everything becomes much clearer after realigning the matrices. For a pizza and its two corresponding principal embedding / unembedding dimensions, $W'_E[a] + W'_E[b] \approx \cos(w_k(a-b)/2) \cdot (\cos(w_k(a+b)/2), \sin(w_k(a+b)/2))$ will be mapped by realigned $W_1$ into its corresponding columns (which are different for every pizza), added with $b_1$ and apply ReLU. The result will then be mapped by the realigned $W_2$, added with *realigned* $b_2$, and finally multipled by $(\cos(w_kc), \sin(w_kc))$.

For the first two principal dimensions, realigned $W_1$ has $44$ corresponding columns (with coefficients of absolute value $> 0.1$). Let the embedded input be $(x, y) = W'_E[a] + W'_E[b] \approx \cos(w_k(a-b)/2) \cdot (\cos(w_k(a+b)/2), \sin(w_k(a+b)/2))$, the intermediate columns are

ReLU($[0.530x - 1.135y + 0.253, -0.164x - 1.100y + 0.205, 1.210x - 0.370y + 0.198, -0.478x - 1.072y + 0.215, -1.017x + 0.799y + 0.249, 0.342x - 0.048y + 0.085, 1.149x - 0.598y + 0.212, -0.443x + 1.336y + 0.159, -1.580x - 0.000y + 0.131, -1.463x + 0.410y + 0.178, 1.038x + 0.905y + 0.190, 0.568x + 1.188y + 0.128, 0.235x - 1.337y + 0.164, -1.180x + 1.052y + 0.139, -0.173x + 0.918y + 0.148, -0.200x + 1.060y + 0.173, -1.342x + 0.390y + 0.256, 0.105x - 1.246y + 0.209, 0.115x + 1.293y + 0.197, 0.252x + 1.247y + 0.140, -0.493x + 1.252y + 0.213, 1.120x + 0.262y + 0.239, 0.668x + 1.096y + 0.205, -0.487x - 1.302y + 0.145, 1.134x - 0.862y + 0.273, 1.143x + 0.435y + 0.171, -1.285x - 0.644y + 0.142, -1.454x - 0.285y + 0.218, -0.924x + 1.068y + 0.145, -0.401x + 0.167y + 0.106, -0.411x - 1.389y + 0.249, 1.422x - 0.117y + 0.227, -0.859x - 0.778y + 0.121, -0.528x - 0.216y + 0.097, -0.884x - 0.724y + 0.171, 1.193x + 0.724y + 0.131, 1.086x + 0.667y + 0.218, 0.402x + 1.240y + 0.213, 1.069x - 0.903y + 0.120, 0.506x - 1.042y + 0.153, 1.404x - 0.064y + 0.152, 0.696x - 1.249y + 0.199, -0.752x - 0.880y + 0.106, -0.956x - 0.581y + 0.223]$).

For the first principal unembedding dimension, it will be taken dot product with

$[1.326, 0.179, 0.142, -0.458, 1.101, -0.083, 0.621, 1.255, -0.709, 0.123, -1.346, -0.571, 1.016, 1.337, 0.732, 0.839, 0.129, 0.804, 0.377, 0.078, 1.322, -1.021, -0.799, -0.339, 1.117, -1.162, -1.423, -1.157, 1.363, 0.156, -0.165, -0.451, -1.101, -0.572, -1.180, -1.386, -1.346, -0.226, 1.091, 1.159, -0.524, 1.441, -0.949, -1.248]$.

Call this function $f(x, y)$. When we plug in $x = \cos(t), y = \sin(t)$, we get a function that well-approximated $8\cos(2t + 2)$ (Figure 26). Therefore, let $t = w_k(a + b)/2$, the dot product will be approximately $8|\cos(w_k(a-b)/2)|\cos(w_k(a+b) + 2)$, or $|\cos(w_k(a-b)/2)|\cos(w_k(a+b))$ if we ignore the phase and scaling. This completes the picture we described above.

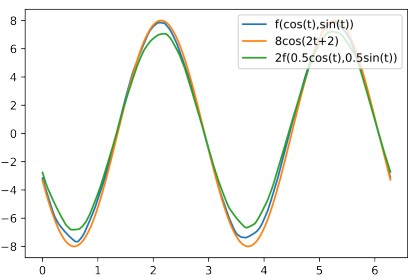

Figure 26: $f(\cos(t), \sin(t))$ well-approximates $8\cos(2t + 2)$.

