# OpenReview forum: "The Clock and the Pizza: Two Stories in Mechanistic Explanation of Neural Networks"
_NeurIPS.cc/2023/Conference — NeurIPS 2023 oral_

### Official Review · Reviewer_DtSC · 2023-06-28

**Soundness:** 3 good
**Presentation:** 4 excellent
**Contribution:** 3 good
**Rating:** 7
**Confidence:** 3

**Summary:**

This paper takes a closer look at the mechanistic explanation of neural networks learning to perform modular addition expanding on recent work that argued that such networks discover a simple "clock" algorithm. The authors demonstrate that changes in initialization and hyperparameters can lead to the discovery of qualitatively different algorithms - most notably what is referred to here as the "pizza" algorithm. This provides evidence that even the simple learning problem of modular addition leads to the discovery of diverse solutions in neural networks and mechanistic explanation requires a more complex analysis.

**Strengths:**

The paper is well written and succeeds in clearly communicating the findings on an active topic in the field of mechanistic interpretability. The analysis is carefully conducted, empirical results are mostly convincing and the conclusion is of importance for the broader field.

**Weaknesses:**

One of the main claims of the paper is that "some networks very similar to the ones trained by [1] preferentially implement a qualitatively different approach" but most of the evidence presented for such different solutions only apply to models that (transiently) remove the attention mechanism.

**Questions:**

1. Focusing on one of the main claims that "some networks very similar to the ones trained by [1] preferentially implement a qualitatively different approach": What is the fraction of neural networks with attention rate 1 that according to your metrics implement the "pizza" algorithm? Judging from Figure 6 it seems to me that this claim might be too strong if the originally investigated model with full attention almost always discovers "clock" solutions.
2. You state that you also find "non-circular algorithms" in your trained networks. What fraction of trained models is non-circular and removed from the main analysis? Again this would be especially insightful to understand in dependence of the attention rate.
3. How are Distance Irrelevance and Gradient Symmetricity related to each other? From my understanding they both intend to measure the same property (pizza vs clock). A scatter plot showing one vs. the other might give some insight on their relationship.
4. In cases where the metrics contradict each other (judging from Figure 6 this sometimes happens), can you still make a confident statement on what algorithm such solutions implement?
6. Could you elaborate how you come to the conclusion that some solutions "implement multiple, imperfect copies of either the Clock or Pizza algorithm in parallel."? How do such solutions work?
7. Could you elaborate why accompanied pizzas achieve almost perfect accuracy (footnote of page 6) despite the failure mode of antipodal pairs?
8. In section 3.4 you conjecture that "accompanying pizzas" are primarily used early in training. Would it be possible to test this hypothesis by comparing the accuracy of "accompanied pizzas" early in training with and without the hypothesised "accompanying pizzas"?
9. Minor point: The word "symmetricity" is unfamiliar to me. Is there a reason to deviate from the more common term symmetry, i.e. calling your metric "Gradient Symmetry"?

**Limitations:**

Limitations have been addressed appropriately.

---

> ### Author Rebuttal · Authors · 2023-08-10
>
> We would like to thank you for your helpful and constructive questions/suggestions! Below is our reply to your questions:
>
> > Q1: The argument “some networks very similar to the ones trained by [1] preferentially implement a qualitatively different approach" seems too strong. What is the fraction of neural networks with attention rate 1 that according to your metrics implement the "pizza" algorithm?
>
> A1: By “similar” we mean structurally similar: our setup is almost identical to [1] except for the introduction of attention rate. From the data we have it seems quite unlikely for a network with attention rate near 1 to implement the pizza algorithm, although we did observe many non-circular (and thus unlikely clock) ones (Fig 6, Appendix C Fig 9). We agree it is somewhat misleading and we will tone down the claim in the updated version.
>
>
> > Q2: What fraction of trained models is non-circular?
>
> A2: For our trained 1,2,3,4-layer 128-width models, the circular (circularity >= 99.5%) ones are 34.31%, 9.95%, 11.55% and 6.08%, respectively. We also attached the circular rate at each attention rate decile and at each width range (figure b & c in the attached pdf).
>
>
> > Q3: What’s the relation between distance irrelevance vs gradient symmetricity?
>
> A3: We consider distance irrelevance as the deciding factor of pizza, as there seem to be limited other reasons for the output logits to depend on the distance. Gradient symmetricity is mostly used to rule out the clock algorithm as the clock algorithm requires multiplying (transformed) inputs, which will result in asymmetric gradients. Following your suggestion, we compiled the scatterplot of distance irrelevance vs gradient symmetricity over all the standard structure experiments we’ve done, and we can indeed see at low distance irrelevance (suggesting pizza) the gradient symmetricity is always close to 1 (suggesting non-clock) except for a few outliers (figure a in the attached pdf).
>
> > Q4: When two metrics contradict, how to make a confident statement?
>
> A4: Following the answer of Q3, we consider distance irrelevance as the defining signature of the Pizza algorithm,  while gradient symmetricity is additional evidence against Clock.
>
> > Q5: What’s the evidence for some solutions "implement multiple, imperfect copies of either the Clock or Pizza algorithm in parallel."?
>
> A5: We agree that it is a bit confusing, but here we refer to the algorithms operating on a single circle as the clock or pizza algorithm, and they are imperfect (the pizza algorithm suffers from antipodal pairs; keeping only the first circle in Model A gives only 32.8% accuracy (L135)). We will clarify this in the revision.
>
> > Q6: Why do accompanied pizzas achieve almost perfect accuracy despite the failure mode of antipodal pairs?
>
> A6: Mechanically, the numbers are arranged differently in each circle so they have different antipodal pairs. In circle #2 of Fig 4, 0 and 10 are roughly antipodal, so circle #2 alone might not be able to get the input (0,10) correct, but we can see that they are relatively close in circle #1, and circle #1 is likely to provide the correct answer. In other words, the multiple copies of the pizza algorithm error correct each other. The accompanying pizzas can also be helpful (Appendix D) although the three circles alone are enough to get close to 100% accuracy.
>
> > Q7: Test the hypothesis that “accompanying pizzas are primarily used early in training”.
>
> A7: We observed the early emergence of a pattern similar to accompanying pizza in training runs (figure f in attached pdf) and removing that circle brings accuracy down from 99.7% to 97.9%. They are less helpful later in the network (removing accompanying pizzas in trained Model A only brings accuracy down to 99.7%).
>
> > Q8: Why use the terminology “symmetricity” rather than “symmetry”?
>
> A8: We think “symmetry” is mostly used as a binary adjective (something either poses symmetry or not), so we used the word “symmetricity” to emphasize the continuous aspect of our metric.

---

> > ### Comment · Reviewer_DtSC · 2023-08-16
> >
> > Thank you for your comprehensive clarifications and the additional experiments/figures to support them. I am happy to increase my score accordingly.

---

### Official Review · Reviewer_7M7r · 2023-07-01

**Soundness:** 4 excellent
**Presentation:** 2 fair
**Contribution:** 3 good
**Rating:** 7
**Confidence:** 4

**Summary:**

In this paper the authors study neural networks doing modular addition of integers, i.e. output $= mod_P( a + b)$ for fixed integer P and input integers a and b. Previous work on these networks has found that small transformers implement a simple _clock_ algorithm. This work verifies this, but shows that if you simplify the transformer’s attention and make your network more like a simple feedforward ReLU network, then you find the networks implement a completely different algorithm they call the _pizza_ algorithm. Their evidence for this includes (i) strange patterns in the correct logit outputs, (ii) gradients that do not fit the clock model but can be understood in their pizza model, (iii) patterns in the logits when the inputs are restricted to particular 2D planes, and (iv) the need for error correcting that leads to particular patterns in a pizza embedding. They then study an algorithmic phase change from pizza to clock as you vary how much attention is included.

**Strengths:**

I thought the large-scale motivation for this work was justified, previous people have claimed that this specific task leads many networks to solve it in the same way. Turns out that isn’t true. That seems important.
I thought the main discovery of this work was very interesting.
I thought the evidence to back up the claim was convincing.
I thought the experiments performed were pretty thorough.
For the most part the claims were not overblown (for example I really appreciated the discussion of non-circular algorithms for solving this task)
Finally the appendix was a bit of a treasure trove.


**Weaknesses:**

I think the paper’s clarity could definitely improve. At times, mainly in the appendix where a lot of the explanation gets relegated, the paper felt rushed and the explanations were terse, expecting a lot from the reader.

I found this especially true of the equations in the appendix, for example:

1.	In appendix A you introduce $s$ on the first line, which is the same as $E_{ab}$ I think. Why do you introduce a new symbol? Why do you not make it clear it is the same as $E_{ab}$ in the main paper?
2.	You then do the same, again in Appendix A, when introducing the symbol $P_c$, which is I believe $Q_{abc}$ from the main paper? You also say thus, which in maths makes me expect to understand that you have derived a result, instead at the moment it reads like a definition of $P_c$ so the use of thus is confusing.
3.	Line 392, in step 1 you talk about an accompanied pizza in a section that readers are expected to reach long before they read about accompanied pizzas where that adds no value as far as I can tell, but just makes the whole thing confusing.
4.	Appendix G was a minefield of strange notational choices in my mind. The layer index was initially denoted with i, but then changes to t, and i is reused as the token index.  I found this needlessly confusing.
5.	You define $x^j$ as the value of the residual stream after j layers, but then talk about $x_{i}^j$ for a couple of lines of the algorithm, before dropping the lower index. I think this is because there’s only one output logit so after the attention you can drop which token the input is coming from, but I still found it confusing at first (because you never say what I just wrote). It felt important to specify that the lower index of x is tokens for understanding the sum over k in the constant attention equation.
6.	For the next description you switch notation but still use x, but now without the top index rather than the bottom one, and use y and z. Could you highlight what is changing?

And beyond the equations I thought that occasionally the appendix was a tough read. For example in appendix H you talked about adding an equal sign. I eventually looked at the caption for figure 19 and understood what that meant, but I thought I’d missed some previously discussed equals sign (I now realise this is a hangover from the original clocks paper). Making sure the writing doesn’t have these kind of moments when the reader hasn’t been told about something and isn’t completely sure where to find out about it seems good. Perhaps you could edit the writing and captions when you try and re-read the paper with fresh eyes to see what is confusing - or get some fresh eyes on it.

A few of small things I am confused by:

1.	I think the sine and cosine addition formulae you are using in step 2 of the algorithm in appendix A, during the development of alpha and beta, are missing factors of 2. [since you use them so often maybe it would be good to state the sine and cosine addition formulae somewhere]
2.	In section 3.4 it says the condition for there to be no antipodal pairs is for p to be prime, isn’t it for p to be odd? What am I missing?
3.	The plots in figure 6 (and all figures like it in the appendix) are titled wrong.
4.	In the formula for distance irrelevance i on the top row should i be a member of $\mathbb{Z}_P$ not $\mathbb{Z}_P^2$?

It seems there are a class of pizza like algorithms (e.g. the two in appendix A), and the evidence listed does not distinguish them. Is this true? Do you know which is happening? If not, perhaps figure 1 is misleading. Instead your claim is that step 3 is one example of a class of pizza type algorithms that are being implemented, and perhaps figure 1 could say that?

**Questions:**


1.	Why do you compute the gradients after having projected to first 6 principal components of the embedding space? What happens if you don’t do this?
2.	Further, why on earth did you think to make this gradient symmetry plot?! Did you already have the pizza algorithm in mind and knew this would distinguish them???
3.	In table 1 it says non-circular algorithms show gradient symmetry, but in general (figure 9) it appears like they don’t. Why does it say this?
4.	The phase transition appears to happen at slightly different points when measured by distance irrelevance or by gradient symmetry, is this true? This definitely seems true in the 2D phase plots. How do you interpret this?
5.	This study restricts itself to circular algorithms, what proportion of models were circular?
6.	The phase change is not discrete, i.e. it does not just jump from the most pizza-ey to the most clock-ey. This could be outside the scope, but what do you think happens in between? An algorithm that is a mixture of the two somehow? Or an output that is a mixture of both algorithms running concurrently? Any evidence in any direction?


**Limitations:**

The authors did a good job discussing limitations.

---

> ### Author Rebuttal · Authors · 2023-08-10
>
> We would like to thank you for your helpful and constructive questions/suggestions! Below is our reply to your questions:
>
> > Q1: Are $E_{ab}$ and $s$ the same?
>
> > Q2: Are $P_{c}$ and $Q_{abc}$ the same?
>
> A2: Thanks for pointing them out! Yes to both - Appendix A was following an older set of notations. We will make sure to correct these mistakes in the final version.
>
> > Q3: Line 392, an accompanied pizza is mentioned but is only defined long after.
>
> A3: Agreed. We will remove the first sentence in the final version.
>
> > Q4: The notations in Appendix G are confusing.
>
> A4: Thanks for the feedback. We have changed all i’s in the first paragraph to t’s.
>
> > Q5: Are $x^j$ and $x^j_{i}$ the same?
>
> A5: Here $x^j$ stands for the whole residual stream vector and $_{i}$ stands for taking its i’th element / dimension. When we drop the lower index, we are performing vector operations on the whole vector $x^j$. We agree it is very confusing and we will make sure to explain the notation choice.
>
> > Q6: Why the change from x to y,z ?
>
> A6: In our particular case there is only one layer, so we feel using x^0 and x^1 will be more confusing so we switched to x and z instead. Except for the notation choice and the expanded loop, nothing is changed.
>
> > Q7: The sine and cosine formula is missing a factor 2. Also specify the formula somewhere.
>
> A7: Great catch! We have added the missing factor and added the two relevant trigonometry formulas at the beginning of Appendix A.
>
>
> > Q8: No antipodal pairs only require p to be odd, not necessarily prime.
>
> A8: Our intention was to stress the case where p is an odd prime which is the most typical setup, but it is indeed confusing. We will change it to be p odd.
>
> > Q9: Plots in Figure 6 (and similar figures in the appendix) are titled wrong.
>
> A9: If we understood your concerns correctly, the plots are not titled (and are labeled) and the text on top are descriptions for the color bar. We agree it is a bit confusing and we will left-align the text to make it clearer. We will update it in the next version.
>
> > Q10: $\mathbf{Z}_p$ or $\mathbf{Z}_p^3$?
>
> A10: It should be $\mathbf{Z}_p$, thanks for pointing that out! We will surely correct it.
>
> > Q11: It seems there are a class of pizza-like algorithms (e.g. the two in appendix A), and the evidence listed does not distinguish them.
>
> A11: Indeed there exists a class of pizza algorithms. The pizza algorithms can differ in how the terms $\cos(w_k(a+b))$ and $\sin(w_k(a+b))$ are approximated by ReLU neurons (Figure 7 and Step 2 in Appendix A). More active neurons could lead to better approximation, but different random seeds and/or hyperparameters may lead to different numbers of active neurons. We will follow your suggestion to update Figure 1 such that it can encompass the whole pizza family and use the current algorithm as a possible special case.
>
> > Q12: Why compute gradients after projecting onto the first six principal components?
>
> A12: The gradient *a*symmetricity is more prominent for the first principal components, as these are more important for the function, and being symmetric is likely easier for the network, and we choose six to be consistent with the later discussion on the three circles. In fact, in the later calculation of gradient symmetricity (Def 4.1) no translation to the principal component space is performed. We’ve attached the same figure with more principal components and without the projection (figure e in the attached pdf). We can see that the gradients are most symmetric for the later principal components as they are not very useful for the algorithm, and without the projection step, the gradient asymmetricity is, in fact, more pronounced for Model B, as the asymmetric gradients on the few principal components are now pronounced across multiple dimensions.
>
> > Q13: What's the rationale behind the gradient symmetry plot?
>
> A13: Past work has shown that neural networks (especially without attention) struggle to learn how to multiply inputs. In this respect, the Clock algorithm felt *unnatural* and we suspected there might be an alternative Pizza-like solution based on linear combination instead.
>
> > Q14: In table 1 it says non-circular algorithms show gradient symmetry, but in general (figure 9) it appears like they don’t.
>
> A14: Thanks for pointing that out! There indeed exist both gradient symmetric and gradient asymmetric non-circular algorithms. We will modify Table 1.
>
> > Q15: The phase transition points seem different when measured by distance irrelevance or gradient symmetricity.
>
> A15: Yes. In short, we believe this is caused by algorithms that are neither clock nor pizza. We consider distance irrelevance to be the *defining* feature of Pizza. Gradient symmetricity is mainly presented as supplementary evidence against the Clock algorithm, which requires multiplying (transformed) inputs, which will result in asymmetric gradients. From figure a in the attached pdf we can see that having low distance irrelevance is indeed a stronger condition than having high gradient symmetricity, suggesting the existence of algorithms that are not clock (high gradient symmetricity) and not pizza (high distance irrelevance).
>
> > Q16: What proportion of models are non-circular?
>
> A16: For our trained 1,2,3,4-layer 128-width models, the circular (circularity >= 99.5%) ones are 34.31%, 9.95%, 11.55% and 6.08%, respectively. We also attached the circular rate at each attention rate decile and at each width range (figure b & c in the attached pdf).
>
> > Q17: The phase change is continuous. What happens in between?
>
> A17: We conjecture that it is a hybrid of clock and pizza: the Algorithm in Appendix 1 takes the dot product of $(\alpha,\beta)$ with $(\cos(w_k c),\sin(w_k c))$ - same as in the clock algorithm. Therefore, it is possible to have some PCA circles operating as the clock algorithm and some operating as the pizza algorithm, and their results are added together before the final dot product with $(\cos(w_k c),\sin(w_k c))$.

---

> > ### Comment · Reviewer_7M7r · 2023-08-13
> > **Reviewer Response**
> >
> > I thank the authors for answering many of my concerns. I had not thought of combining different circles each doing a different algorithm, that is an interesting possibility, and the information on proportion of networks that are circular is interesting.
> >
> > I realise I was being an idiot about figure 6 since, as you say, I indeed thought the title of the colourbar was the title of the plot. So you can ignore that!
> >
> > I think my scoring of the work still stands, and hope to see a cleaned up version of the paper accepted.

---

### Official Review · Reviewer_HMiA · 2023-07-06

**Soundness:** 4 excellent
**Presentation:** 4 excellent
**Contribution:** 4 excellent
**Rating:** 7
**Confidence:** 3

**Summary:**

*Background*: Previous work in mechanistic interpretability has identified a particular algorithm that NNs implement (the clock algorithm) to solve modular addition. However, under certain architecture changes, the authors notice that NNs implement a different algorithm.

The main goal of this paper is to present inconsistencies in the clock algorithm for neural networks without attention (ie: an inductive bias that allows the model to implement multiplication) and motivate a different algorithm (the pizza) that explains a different algorithm through which such networks learn modular attention. The observations are backed by experiments on linearly interpolating between a NN with attention and an NN without it. The experiments also demonstrate that a single algorithm doesn’t always win: different models can and do ensemble multiple copies of both algorithms in parallel.

**Strengths:**

The paper is very well written and the proofs and arguments presented seem airtight.  Most of the questions I’ve had while reading the paper are either addressed in subsequent sections or in the appendix. The experiments are simple yet, I believe, comprehensive in evaluating the arguments presented. This is also an exciting emerging area of research and should lead to interesting discussions in the mechanistic interpretability community.

**Weaknesses:**

This work raises a lot of interesting questions, but I really can’t find any egregious logical inconsistencies with this work.
meta-(non)concern: This work largely relies on a problem from a paper that hasn’t been peer-reviewed. However, I do not think this is a reason to reject this work.

Overall, I recommend _acceptance_.

**Questions:**

* L78: Why the choice of 6 vectors specifically? For exposition? Will choosing less significant components introduce unnecessary noise?
* L113: What would happen with an odd number of ReLU units?


**Limitations:**

The authors have addressed limitations in the manuscript.

---

> ### Author Rebuttal · Authors · 2023-08-10
>
> We would like to thank you for your helpful and constructive questions/suggestions! Below is our reply to your questions:
>
> > Q1: L78: Why the choice of 6 vectors specifically? For exposition? Will choosing less significant components introduce unnecessary noise?
>
> A1: The main motivation is that the gradient *a*symmetricity is more prominent for the first principal components, as these are more important for the function, and being symmetric is likely easier for the network. The choice also helps to be more consistent with the later discussions on 3 circles (Fig 4, Fig 5), which correspond to the first 6 principal components. In fact, in the later calculation of gradient symmetricity (Def 4.1) no translation to the principal component space is performed. We’ve attached the same figure with more (20) principal components and without the principal component projection (figure e in attached pdf).
>
> > Q2: L113: What would happen with an odd number of ReLU units?
>
> A2: We can implement absolute value $|x|$ by $\text{ReLU}(x)-\text{ReLU}(-x)$. If there are an odd number of ReLU units, some could be dead neurons (in the sense that the activation is near-zero for all inputs). There are also multiple possible variants of the pizza algorithm (Appendix A).

---

> > ### Comment · Reviewer_HMiA · 2023-08-13
> >
> > Thank you for the clarifications! I'm keeping my score where it is.

---

### Official Review · Reviewer_ZmSF · 2023-07-15

**Soundness:** 4 excellent
**Presentation:** 4 excellent
**Contribution:** 3 good
**Rating:** 8
**Confidence:** 4

**Summary:**

The authors present a novel algorithm as a mechanistic explanation of neural networks for modular addition. It is noted that the model without attention fails to implement the ‘Clock’ algorithm. This assertion is substantiated with evidence related to gradient symmetricity and logit patterns. The authors then propose an alternative solution, named the ‘Pizza’ algorithm, supported by evidence concerning logit patterns via circle isolation and accompanying ‘pizza’. Ultimately, they demonstrate the presence of an algorithmic phase transition along the attention rate and model width, employing metrics that indicate gradient symmetricity and distance irrelevance.


**Strengths:**

The paper is well-structured and supports its arguments with solid experiments. The authors demonstrate that a neural network is capable of learning diverse algorithms for the same task. They introduce an impressive procedure for interpreting the neural network via embedding vectors. This methodology has the potential for extension to more complex models and tasks.

**Weaknesses:**

The authors employ the term logit $Q_{abc}$ as well as the term output logit, which refers to the un-normalized log probability. The choice of terminology, however, proves to be confusing. Given that $Q_{abc}$ is not used in the model and is a concept introduced by the authors themselves, it would be beneficial to rename $Q_{abc}$ to a more intuitive term like "value" or "rank".

**Questions:**

What prevents us from directly ascertaining the algorithm employed by the model? Couldn't it be possible to determine the intermediate vector $E_{ab}$ to gain insights into the algorithmic process?

**Limitations:**

As the authors have noted, their focus lies on a simple learning problem. Significant further work is required to adapt their techniques for use with the more complex models typically employed in real-world tasks.

---

> ### Author Rebuttal · Authors · 2023-08-10
>
> We would like to thank you for your helpful and constructive questions/suggestions! Below is our reply to your questions:
>
> > Q1: Calling $Q_{abc}$ output logit is confusing.
>
> A1: This terminology has been used in many previous interpretability studies [Nanda2023] [Wang2023], so we are using standard nomenclature in this research area.
>
>
>
> > Q2: Can we determine the intermediate vector $E_{ab}$ to gain insights into the algorithmic process?
>
> A2: It is certainly possible, and we can prove that mechanistically in constant-attention Transformers, the computation starts by adding two embeddings (Appendix G).
>
>
> **Reference**
>
> [Nanda2023] “Progress measures for grokking via mechanistic interpretability”, Nanda et al.
>
> [Wang2023] “Interpretability in the Wild: a Circuit for Indirect Object Identification in GPT-2 small”, Wang et al.

---

> > ### Comment · Reviewer_ZmSF · 2023-08-14
> > **Reviewer Response**
> >
> > Thank you for your clarification! I'll maintain my rating.

---

### Official Review · Reviewer_wANB · 2023-07-19

**Soundness:** 3 good
**Presentation:** 3 good
**Contribution:** 3 good
**Rating:** 7
**Confidence:** 4

**Summary:**

In this work, the authors focus on the problem of learning modular addition in NNs. Using clock and pizza algorithms, they show that model exhibits sharp algorithm transitions, which are affected by layer width and attention strengths, often resulting in the parallel occurrence of these phases.  A series of experiments are performed on the single-layer network to support the hypothesis proposed in this work.

**Strengths:**

1. Well-written paper.
2, an useful contribution in terms of interpretability of NNs.
3. Novel contribution in terms of analyzing the algorithmic transitions.

**Weaknesses:**

1. Ablation study is missing
2. Did authors perform a grid search to select best hyper-parameters? Then that should be mentioned with ranges in the appendix.




**Questions:**

In terms of interpretability there are models inspired by formal language theory that insert rules [1,3] and extract rules [2-7], termed as interpretable by extractions and such models are even tested on mathematical reasoning[4] task and are known to be turing complete even with finite precision and time [8]. Authors should discuss this line of work, as they are relevant.

How to determine the threshold for the circularity, does value >= 99.5% works for all model/architectures in terms of thresholding for circularity. It would be ideal if authors can provide empirical bound for this and how to determine such threshold.

Can this framework be extended to convolutions with tensor weights or stateful models such as RNNs?

Authors do point out that deeper models leads to non-circular algorithms, but what is the bound for that? After how many layers non-circular behavior is shown by various models? At minimum providing empirical results will further strengthen this work. I would also like to see some empirical bounds on attention rate, what quantifies as high attention rate and what is low attention rate. Can author provide ablation study on various value of attention rate and also the width of layers?

Few additional comments that are not clear from the manuscript.

How does the model effectively interpolate between the memorizing and generalizing solutions?

Does this also work with sinusoidal embeddings, masked embedding? Does the choice of embedding have an issue in generalization?

The authors do mention pruning the weights, so what effect does sparsity have in the model performance? Can authors comment on this? Like how the two-phase switch?





Finally I would like to see total computational time required by the model including FLOPS and also standard error for various trials on proposed experiments.

Minor comments
The figure 4 should be improved, its difficult to read values on y-axis and also values overlap in the circular diagram.  Same goes for other figures too.

1.	Omlin, C.W. and Giles, C.L., 1996. Rule revision with recurrent neural networks. IEEE Transactions on Knowledge and Data Engineering, 8(1), pp.183-188.
2.	Tiňo, P. and Šajda, J., 1995. Learning and extracting initial mealy automata with a modular neural network model. Neural Computation, 7(4), pp.822-844.
3.	Mali, A.A., Ororbia II, A.G. and Giles, C.L., 2020. A neural state pushdown automata. IEEE Transactions on Artificial Intelligence, 1(3), pp.193-205.
4.	Mali, A., Ororbia, A.G., Kifer, D. and Giles, C.L., 2021, May. Recognizing and verifying mathematical equations using multiplicative differential neural units. In Proceedings of the AAAI Conference on Artificial Intelligence (Vol. 35, No. 6, pp. 5006-5015).
5.	Weiss, G., Goldberg, Y. and Yahav, E., 2018, July. Extracting automata from recurrent neural networks using queries and counterexamples. In International Conference on Machine Learning (pp. 5247-5256). PMLR.
6.	Wang, C., Lawrence, C. and Niepert, M., 2022. State-Regularized Recurrent Neural Networks to Extract Automata and Explain Predictions. IEEE Transactions on Pattern Analysis and Machine Intelligence, 45(6), pp.7739-7750.
7.	Okudono, T., Waga, M., Sekiyama, T. and Hasuo, I., 2020, April. Weighted automata extraction from recurrent neural networks via regression on state spaces. In Proceedings of the AAAI Conference on Artificial Intelligence (Vol. 34, No. 04, pp. 5306-5314).
8.	Stogin, J., Mali, A. and Giles, C.L., 2020. A provably stable neural network Turing Machine. arXiv preprint arXiv:2006.03651.

**Limitations:**

As highlighted above, the main point is an ablation study to support the hypothesis and computational overhead.

**********
Score increased after Author rebuttal Responses

---

> ### Author Rebuttal · Authors · 2023-08-10
>
> We would like to thank you for your helpful and constructive questions/suggestions! Below is our reply to your questions:
>
> > Q1: Ablation study is missing. Can authors provide ablation studies on various values of attention rate and also the width of layers?
>
> A1: Thank you for the suggestion! If we understand correctly, the ablation study you are requesting is already provided in Figure 6 of the submission, which shows behavior at  various values of attention rate and model width.
>
> > Q2: Did authors perform a grid search to select the best hyper-parameters? Then that should be mentioned with ranges in the appendix.
>
> A2: No. We largely followed [Nanda2023] for hyperparameter setups and we chose p=59 following [Liu2023] to simplify the investigation.
>
>
> > Q3: Can you discuss the literature and cite relevant papers?
>
> A3: Thanks for pointing us to these references, which are indeed relevant to our work. We would like to include the citations in the next updated version.
>
>
> > Q4: How to determine the threshold for circularity?
>
> A4: This is necessarily a bit subjective. Circularity is fundamentally a qualitative phenomenon, and this threshold indicates how tolerant we are in terms of considering some shapes as being approximately circular (which the authors all agreed was true of shapes with circularity >= 99.5%).
>
>
> > Q5: Can this analysis apply to convolutional neural networks or recurrent neural networks?
>
> A5: Yes, our analysis can apply to other architectures, e.gCNN and RNN. This analysis does not require inspecting latent representations (which are specific to architectures), only involving output logits and gradients wrt to input embeddings (which are universal to all architectures).
>
> > Q6: For the transition from circular to non-circular algorithms, what is the bound (phase transition point) along depth, attention rate, width?
>
> A6: There is no clear phase transition from circular to non-circular algorithms against attention rate and width, but depth 1 networks are clearly more likely to be circular solutions than deeper (2-4 layers) networks. See figure b and c in the attached pdf.
>
>
>
> > Q7: How does the model effectively interpolate between the memorizing and generalization solutions?
>
> A7: Our work focuses mostly on analyzing the final (generalization) solution. The training dynamics (how the model interpolates from a memorizing to a generalization solution) is interesting but might be out of the scope of this work.
>
>
> > Q8: Does this also work with sinusoidal embeddings, masked embedding? Does the choice of embedding have an issue in generalization?
>
> A8: Yes, we believe other types of embeddings can also lead to generalization. We’re currently using learnable positional embedding, but our proposed pizza algorithms do not depend on them, so we believe the pizza algorithms also exist under sinusoidal positional embeddings. As for masked embeddings, in the 1-layer case masked embedding is equivalent to bidirectional embedding (since attention to token #2 to token #1 doesn’t count) so our conclusions should remain valid.
>
> > Q9: How does pruning weights (hence sparsity) affect performance?
>
> A9: It is unclear to us whether our analysis has implications for sparsity. Norm-wise strong concentration is observed and its relationship with attention rate, distance irrelevance is observed but weak (figure g in the attached pdf). We observed some difference in parameter distribution for Model A and Model B (figure d the attached pdf) and we believe it is primarily a result of different model configuration (for example, query and key matrices are ignored by constant-attention Transformers).
>
>
> > Q10: Can you provide computation time (including FLOPs) and error bars?
>
> Q10: We spent roughly 226 GPU days on a V100 cluster with ~30% utilize rate, so the total computation is around 4e19 FLOPs.
> It is hard to provide an error estimation since we are sampling with respect to multiple parameters, but we have made the full distribution available.
>
>
> > Q11: Figure 4 (and so as other figures) should be improved to maximize readability.
>
> A11: Thanks for the suggestion! We will increase fonts and do other optimization if needed.
>
>
> **References**
>
> [Nanda2023] “Progress measures for grokking via mechanistic interpretability”, Nanda et al.
>
> [Liu2023] “Towards Understanding Grokking: An Effective Theory of Representation Learning”, Liu et al.

---

> ### Comment · Reviewer_wANB · 2023-08-16
> **Rebuttal Response**
>
> I thank the authors for their detailed responses. I have also read other reviews and responses; thus, I am increasing my score and moving toward acceptance. I believe the updated paper will have all the components promised by the authors in their rebuttal. Overall Good work.

---

### Author Rebuttal · Authors · 2023-08-10

We would like to thank all reviewers for their helpful and constructive suggestions, which will greatly improve the final version of the paper. Besides individual responses, we want to summarize our responses/updates to reviewers’ common questions here. Reviews prompted us to try several additional experiments, which have led to fruitful discoveries:

### Distance irrelevance vs gradient symmetricity

Distance irrelevance is a rather surprising and defining feature of the pizza algorithm while the gradient symmetricity is mainly presented as supplementary evidence mostly used to rule out the Clock algorithm, which requires multiplying (transformed) inputs and hence has asymmetric gradients. We plotted the relationship between gradient symmetricity and gradient irrelevance for all the 1~4 layer 128-width models we trained, and we confirmed that low distance irrelevance (suggesting Pizza) almost always implies close to 1 gradient symmetricity (suggesting non-Clock) (figure a in the attached pdf).

### Accompanying pizzas are employed early in training

We observed the early emergence of accompanying pizza in training runs (figure f in the attached pdf; irrelevant principal components not displayed for space concerns). The model was trained 600 epochs at the time and reached 99.7% accuracy on the validation set (for reference, all the models we reported are trained for 20000 epochs). From the logit pattern, the first two principal components of the input embedding resemble the pizza algorithm, and the 13th and 14th principal components resemble the accompanying pizza. It is surprising that although the two components do not exactly resemble a circle due to the lack of training, the logit pattern is still clear and corresponds to the first circle. Removing this “accompanying pizza” brings the accuracy down to 97.9%.

### Projection for gradient symmetricity
We projected the gradients of the models to the principal components so as to match our description of algorithms on principal components. The less significant principal components account less for the correct functioning of the model and we observed their corresponding gradients concentrating near 0 (fig e left in the attached pdf). If we consider the raw unprojected gradients, the asymmetricity of a few model B’s principal components’ gradients is more pronounced as it now affects multiple raw gradient dimensions (fig e right in the attached pdf).

### Circularity with respect to layer, attention rate, and width
We computed the circular rate (circularity >= 99.5%) of models with respect to the number of layers, attention rate, and width (fig b and fig c in the attached pdf). We found out that the circular rate is higher for 1-layer models than multiple-layer models, and among 1-layer models circular rate is higher when the attention rate is closer to 0 or 1. Our explanation is that Pizza and Clock are two circular phases that are easiest to obtain at 1 layer and attention rate 0 or 1, correspondingly, so setups closer to these two phases are more likely to be drawn to them, resulting in similarly circular states.

### Sparsity and norm distribution
We plotted the relationship between attention rate, distance irrelevance, gradient symmetricity, and parameter L2 norm (figure g in the attached pdf). Here the parameter stands for all the trainable coefficients in the trained model. Besides clear concentration, we can see a slight increase in the mean L2 norm as attention rate and distance irrelevance increase. We also observed a slight increase in L2 norm from Model A (22.9) to Model B (24.8). Their parameter distributions are also different (figure d in the attached pdf). We believe this is a result of different attention rate setups. Namely, for Model A with constant attention (attention rate 0), the query and key matrices are ignored so they are optimized to near-0 values.



Besides empirical experiments, we also incorporate other suggestions from reviewers, including clarifying the family of pizza algorithms (Reviewer 7M7r), related work discussions (Reviewer wANB) and writing (Reviewer 7M7r, ZmSF, DtSC).

---

### Decision · Program_Chairs · 2023-09-21

**Decision:**

Accept (oral)

**Comment:**

All reviewers agree that this paper is clearly presented and soundly supported, both mathematically and empirically. There is generally a high level of enthusiasm for accepting it. The particular contribution---showing that one of the main demonstrations of mechanistic interpretability is not robust---is timely, and should influence the thinking of people working in this area.